# AERO: SOFTMAX-ONLY LLMS FOR EFFICIENT PRIVATE INFERENCE

## ABSTRACT

The pervasiveness of proprietary language models has raised privacy concerns for users' sensitive data, emphasizing the need for private inference (PI), where inference is performed directly on encrypted inputs. However, current PI methods face prohibitively higher communication and latency overheads, primarily due to nonlinear operations. In this paper, we present a comprehensive analysis to understand the role of nonlinearities in transformer-based decoder-only language models. We introduce AERO, a four-step architectural optimization framework that refines the existing LLM architecture for efficient PI by systematically removing nonlinearities such as LayerNorm and GELU and reducing FLOPs counts. For the *first time*, we propose a Softmax-only architecture with significantly fewer FLOPs tailored for efficient PI. Furthermore, we devise a novel entropy regularization technique to improve the performance of Softmax-only models. AERO achieves up to $4.23\times$ communication and $1.94\times$ latency reduction. We validate the effectiveness of AERO by benchmarking it against the state-of-the-art.

## 1 INTRODUCTION

**Motivation.** The widespread adoption of proprietary models like ChatGPT Achiam et al. (2023) significantly raised the privacy concerns to protect the users' sensitive (prompt) data Staab et al. (2024); Mireshghallah et al. (2024); Priyanshu et al. (2023); Lauren & Knight (2023), while also preventing the attacks aimed at extracting model weights Carlini et al. (2024); Jovanović et al. (2024).

This emphasizes the need for private inference (PI) where a user sends the encrypted queries to the service provider without revealing their actual inputs, and the inference is performed directly on encrypted inputs, assuring the privacy of input and protection of the model's weight.

Despite their promises, current PI methods remain impractical due to their prohibitive latency and communication overheads—generating a single output token with GPT-2 model (125M parameters) on 128 input tokens takes 8.2 minutes and requires 25.3 GBs communication (Figure 1), scaling to 30.7 minutes and 145.2 GBs for context size of 512 (Table 7). These overheads stem largely from the nonlinear operations, crucial for model performance, in a transformer-based large language model (LLM), such as GELU, LayerNorm, and Softmax Hou et al. (2023); Lu et al. (2025).

**Challenges.** Current PI solutions for transformer-based models (e.g., ViT, BERT) either *neglect* the cost of LayerNorm (Li et al., 2023a; Zeng et al., 2023; Zhang et al., 2023; Chen et al., 2023) or approximate nonlinear operations using polynomial functions Zimerman et al. (2024); Dhyani et al. (2024). Nonetheless, polynomial approximation methods have their limitations: their accuracy is highly sensitive to data-specific initial guesses Knott et al. (2021), and their effectiveness is confined to narrow input ranges Zimerman et al. (2024). Moreover, networks employing higher-degree polynomials for improved approximation precision are notoriously difficult to train and optimize.

Meanwhile, the nonlinearity reduction methods, used for improving plaintext speed, offer *very-limited* potential to improve the PI efficiency. For instance, (He et al., 2023; Noci et al., 2023; He & Hofmann, 2024) has explored architectural heuristics to design LayerNorm-free LLMs; however, their broader implications on the choices of activation function, a key bottleneck in PI, remains largely unexamined.

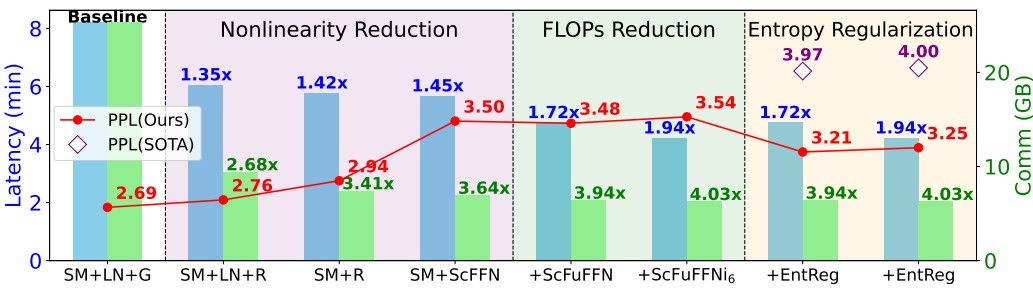

Figure 1: Latency and communication savings through nonlinearity and FLOPs reduction steps when AERO is applied on GPT-2, and trained from scratch on CodeParrot dataset. Further, we benchmark AERO against the SOTA He & Hofmann (2024) at iso-latency points. See Table 4 for a detail analysis.

**Our techniques and insights.** We conducted an in-depth analysis of the role of non-linearities in transformer-based LLMs. Our key findings are: (1) LayerNorm-free models exhibit a preference for ReLU over GELU in feed-forward network (FFN), making them more PI-friendly; and (2) training instability, as entropy collapse in deeper layers, in the Softmax-only model can be prevented by normalizing FFN weights, avoiding the nonlinear computations (unlike LayerNorm) at inference.

We observed a phenomenon we term *entropic overload*, where a disproportionately larger fraction of attention heads stuck at higher, close to their maximum, entropy values in LN-free with GELU, and Softmax-only models. We hypothesize that the entropic overload causes a lack of diversity and specialization in attention heads, squandering the representational capacity of attention heads. This leads to performance degradation, indicated by a higher perplexity.

To mitigate the entropic overload, we propose a novel entropy regularization technique that penalizes the extreme entropy values at training and avoids the deviation from well-behaved entropy distribution.

**Results and implications.** As shown in Figure 1, substituting GELU with ReLUs in the baseline GPT-2 model alone reduces the communication and latency overheads by $2.68\times$ and $1.35\times$, respectively. Eliminating LayerNroms further improves these savings to $3.41\times$ and $1.42\times$. Similar improvements are observed with the Pythia model (see Figure15).

Since the FFN in the Softmax-only model is performing only the linear transformations, merging the linear layers into a single linear layer reduces the FFN FLOPs by $8\times$ and gains significant speedup *without increasing* the perplexity (see Figure 1). Furthermore, our analysis reveals that the linear transformations performed by early FFNs are crucial for training stability in the Softmax-only model, while deeper FFNs can be pruned. This provides additional opportunities for FLOPs reduction.

**Contributions.** Our key contributions are follows:

1. We thoroughly characterize the role of GELU and LayerNorm nonlinearities in transformer-based LLMs by examining their impact on the attention score distribution using Shannon's entropy, offering insights for tailoring existing LLM architectures for efficient PI.
2. We introduced AERO, a four-stage optimization framework, and designed a Softmax-only model with fewer FLOPs, achieving up to **1.94×** speedup and **4.23×** communication reduction.
3. We introduce a novel entropy regularization technique to boost the performance of the Softmax-only model, which achieves **6%** - **8%** improvement in perplexity.
4. We conducted extensive experiments across various context sizes (128, 256, 512) and model depths (12L and 18L) on a wide range of training tokens (1.2B to 4.8B) from the CodeParrot Face and Languini dataset Stanić et al. (2023) on GPT-2 and Pythia Biderman et al. (2023) models.

## 2 PRELIMINARIES

**Notations.** We denote the number of layers as $L$, number of heads as $H$, model dimensionality as $d$, head dimension as $d_k$ (where $d_k = \frac{d}{H}$), and context length as $T$. Table 1 illustrates the abbreviations for architectural configurations with simplified nonlinearities in a transformer-based LLM.

**An overview of transformer-based decoder-only architecture.** A transformer-based LLM is constructed by sequentially stacking $L$ transformer blocks, where each block is composed of two

sub-blocks: an attention mechanism and a feed-forward network (FFN), both having their own residual connections and normalization layers, positioned in the Pre-LN order to improves training stability (Xiong et al., 2020). Formally, transformer blocks take an input sequence $\mathbf{X}_{\text{in}} \in \mathbb{R}^{T \times d}$, consisting of $T$ tokens of dimension $d$, and transform it into $\mathbf{X}_{\text{out}}$ as follows:

$$\mathbf{X}_{\text{out}} = \hat{\mathbf{X}}_{\text{SA}} + \text{FFN}_{\text{GELU}}(\text{LayerNorm}_2(\hat{\mathbf{X}}_{\text{SA}})), \text{ where } \hat{\mathbf{X}}_{\text{SA}} = \mathbf{X}_{\text{in}} + \text{MHA}(\text{LayerNorm}_1(\mathbf{X}_{\text{in}})). \quad (1)$$

The Multi-Head Attention (MHA) sub-block enables input contextualization by sharing information between individual tokens. MHA employs the self-attention mechanism to compute the similarity score of each token with respect to all other tokens in the sequence. In particular, self-attention mechanism transform the input sequence $\mathbf{X}$ into $\mathbf{Attn}(\mathbf{X})$ as follows:

$$\mathbf{Attn}(\mathbf{X}) = \left(\text{Softmax}\left(\frac{1}{\sqrt{d_k}}(\mathbf{X}\mathbf{W}^Q)(\mathbf{X}\mathbf{W}^K)^\top + \mathbf{M}\right)\right)\mathbf{X}\mathbf{W}^V. \quad (2)$$

Here, each token generates query($Q$), key($K$), and value($V$) vectors through the linear transformations $\mathbf{W}^Q, \mathbf{W}^K$, and $\mathbf{W}^V \in \mathbb{R}^{d \times d_h}$, respectively. Then, similarity scores are computed by taking the dot product of the $Q$ and $K$ vectors, scaled by the inverse square root of the $K$ dimension, and passed through a softmax function to obtain the attention weights. These weights are then used to compute a weighted sum of the $V$ vectors, producing the output for each token. For auto-regressive models (e.g., GPT), mask $\mathbf{M} \in \mathbb{R}^{T \times T}$, which has values in $\{0, -\infty\}$ with $\mathbf{M}_{i,j} = 0$ iff $i \geq j$, is deployed to prevent the tokens from obtaining information from future tokens.

The MHA sub-block employs a self-attention mechanism across all the heads, each with its own sets of $Q$, $K$, and $V$. This allows the attention heads to focus on different parts of the input sequence, capturing various aspects of the input data simultaneously. The outputs from all heads are concatenated and linearly transformed ($\mathbf{W}^O \in \mathbb{R}^{d \times d}$) to produce the final MHA output as follows:

$$\text{MHA}(\mathbf{X}) = \text{Concat}\left(\text{Attn}_1(\mathbf{X}), \text{ Attn}_2(\mathbf{X}), \text{ Attn}_3(\mathbf{X}), \ldots, \text{Attn}_H(\mathbf{X})\right)\mathbf{W}^O. \quad (3)$$

Following the MHA sub-block, the FFN sub-block transforms each token independently. The FFN sub-blocks have a single hidden layer whose dimension is a multiple of $d$ (e.g., $4d$ in GPT (Radford et al., 2019) models). Specifically, the FFN sub-block first applies a linear transformation to the input $\mathbf{X}$ using $\mathbf{W}_{\text{in}}^{\text{ffn}} \in \mathbb{R}^{d \times 4d}$, followed by a non-linear transformation using an activation function such as GELU. This is then followed by another linear transformation using $\mathbf{W}_{\text{out}}^{\text{ffn}} \in \mathbb{R}^{4d \times d}$, as follows:

$$\text{FFN}(\mathbf{X}) = (\text{GELU}(\mathbf{X}\mathbf{W}_{\text{in}}^{\text{ffn}}))\mathbf{W}_{\text{out}}^{\text{ffn}} \quad (4)$$

The combination of MHA and FFN sub-blocks, along with residual connections and normalization layers, allows transformer models to learn the contextual relationships between tokens effectively.

**Threat model for private inference.** We consider the standard two-party (2PC) client-server setting used in PPML, which provides security against semi-honest (honest-but-curious) adversaries bounded by probabilistic polynomial time Zhang et al. (2025); Lu et al. (2025); Pang et al. (2024); Hou et al. (2023). Both parties follow protocol specifications but may attempt to gain additional information from their outputs about the other party's input. In this 2PC setting, the server holds the propriety GPT model (e.g., ChatGPT), and the client queries the model with a piece of text (prompt). The protocols ensure that the server does not know anything about the client's input and the output of their queries, and the client does not know anything about the server's model except its architecture.

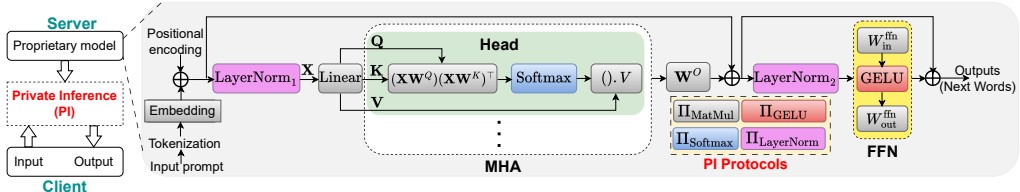

Figure 2: An illustration of threat model and cryptographic protocols used for LLM private inference.

# 3 REMOVING NONLINEARITY IN TRANSFORMER-BASED LLMS

In this section, we investigate the role of non-linearities in the learning dynamics and internal representations of a transformer-based autoregressive decoder-only LLM. We design a controlled

experimental framework that systematically removes non-linear components from the architecture (see Table 1), and trains models from scratch.

Table 1: Architectural configurations of nonlinearities in LLMs, illustrating the combinations of Softmax (SM), LayerNorm (LN), GELU (G), and ReLU (R) functions (see Eq. 1, 2, 3 and 4).

| Abbreviation | Architectural configuration |
|---|---|
| SM + LN + G | $\mathbf{X}_{\text{out}} = \text{FFN}_{\text{GELU}}(\text{LayerNorm}_2(\text{MHA}(\text{Attn}_{\text{Softmax}}(\text{LayerNorm}_1(\mathbf{X}_{\text{in}})))))$ |
| SM + LN + R | $\mathbf{X}_{\text{out}} = \text{FFN}_{\text{ReLU}}(\text{LayerNorm}_2(\text{MHA}(\text{Attn}_{\text{Softmax}}(\text{LayerNorm}_1(\mathbf{X}_{\text{in}})))))$ |
| SM + LN | $\mathbf{X}_{\text{out}} = \text{FFN}_{\text{Identity}}(\text{LayerNorm}_2(\text{MHA}(\text{Attn}_{\text{Softmax}}(\text{LayerNorm}_1(\mathbf{X}_{\text{in}})))))$ |
| SM + G | $\mathbf{X}_{\text{out}} = \text{FFN}_{\text{GELU}}(\text{MHA}(\text{Attn}_{\text{Softmax}}(\mathbf{X}_{\text{in}})))$ |
| SM + R | $\mathbf{X}_{\text{out}} = \text{FFN}_{\text{ReLU}}(\text{MHA}(\text{Attn}_{\text{Softmax}}(\mathbf{X}_{\text{in}})))$ |
| SM | $\mathbf{X}_{\text{out}} = \text{FFN}_{\text{Identity}}(\text{MHA}(\text{Attn}_{\text{Softmax}}(\mathbf{X}_{\text{in}})))$ |

To analyze internal representations, we use Shannon's entropy to examine the impacts of nonlinearities on the attention score distribution (see Appendix A.1 for its justification). We highlight key insights and findings, offering practical guidelines for tailoring LLM architectures for efficient PI.

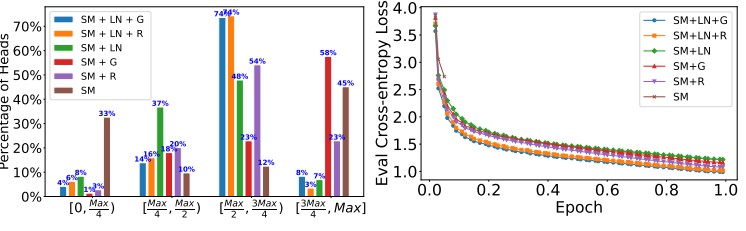

(a) Headwise entropy distribution    (b) Loss curve

| Configurations | PPL | $+\Delta(\%)$ |
|---|---|---|
| SM + LN + G | 2.69 | 0.00 |
| SM + LN + R | 2.76 | 2.53 |
| SM + LN | 3.38 | 25.58 |
| SM + G | 3.20 | 18.92 |
| SM + R | 2.94 | 9.20 |
| SM | NaNs | - |

Table 2: Evaluation perplexity for GPT-2 (small) models with fewer nonlinearities, corresponding to Figure 3b. $\Delta$ is increase in PPL over baseline network.

Figure 3: (a) The fraction of attention heads distributed across different entropy ranges, and (b) evaluation loss for GPT-2 (small) models with fewer nonlinearities, when trained from scratch on CodeParrot dataset.

**Well-behaved entropy distribution** We begin by analyzing the headwise entropy distribution of baseline architecture with GELU and ReLU in the FFN, i.e., configurations SM + LN + G and SM + LN + R respectively. We find that the majority of heads ($\approx$90%) possess entropy values between $\frac{\text{max}}{4}$ and $\frac{3\text{max}}{4}$, where max is maximum observed entropy value among all heads (see Figure 3a). This concentration in the mid-entropy range, while avoiding extremes, demonstrates a well-behaved distribution, providing a benchmark for assessing the impact of nonlinearities on model behavior.

**Entropic overload** We observed that in certain nonlinearity configurations, a disproportionately large fraction of the attention heads exhibit higher entropy values (between $\frac{3\text{max}}{4}$ and max). We term this phenomenon as entropic overload and hypothesize that this imbalance results in *under-utilization* of the network's representational capacity, as too many heads engaged in exploration, hindering the model from effectively leveraging the diversity of attention heads.

To investigate further, we examined how entropy values evolve during training. Typically, all heads start with higher entropy values, indicating an initial exploration phase, and gradually adapt to balance exploration and exploitation in baseline networks (see Figure 12). However, in the absence of certain nonlinearities, this balance is disrupted, preventing attention heads from specializing and refining their focus on critical aspects of the input, thereby diminishing overall performance.

### 3.1 DESIRABLE ACTIVATION FUNCTION IN LAYERNORM-FREE LLMs

We first remove LayerNorm from the LLM architecture and study the desirable activation function in this design, as the absence of LayerNorm can destabilize activation statistics.

**Observation 1: ReLU significantly outperforms GELU in LayerNorm-Free LLMs.** While GELU is typically preferred over ReLU in conventional transformer-based models due to its smooth and differentiable properties that improve performance and optimization, our empirical findings indicate the *opposite trend* for LayerNorm-free models— using ReLU in the FFN exhibit better learning

dynamics than their GELU counterpart. This leads to an **8.2%** improvement in perplexity for GPT-2 (see Figure 3 and Table 2). A similar trend has been observed on the LN-free Pythia-70M model across various context lengths (see Table 8).

To further strengthen our findings, we conducted experiments with a learnable negative slope in the leaky ReLU activation function with two configurations: 1) layer-wise, where each layer has its independent learnable slope, and 2) global, where a single learnable slope is shared across all layers. Results are shown in Figure 4. Interestingly, in the layerwise setting, the early layers initially learn

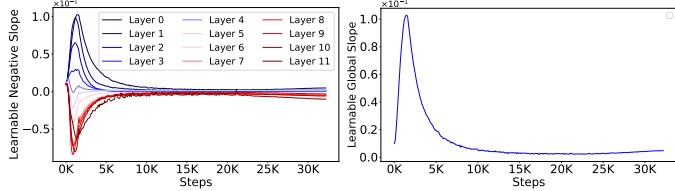

(a) Layerwise learnable slope      (b) Global learnable slope

Figure 4: Learnable negative slope for leaky ReLU in the FFN of LN-free GPT-2. (a) Layerwise slopes and (b) global slope, both converge toward zero during training, indicating a preference for zero negative slope in LN-free architectures.

a positive slope while the deeper layers learn a negative slope. However, as training progresses, all layers converge to a near-zero slope. In the global setting, the slope first shifts to positive before converging to near zero. Refer to Figure 13 for their layerwise entropy dynamics.

This highlights the distinct learning dynamics of nonlinearity choices, and a natural preference for zero negative slope, similar to ReLU, in the FFN activation function of the LN-free model.

**Observation 2: Early layers in the LayerNorm-Free model with GELU in FFN experience entropic overload**. To understand the zero negative slope preference for the FFN activation function in LN-free architecture, we analyzed the headwise entropy values of LN-free models with GELU and ReLU, when trained from scratch, and compared them to their baseline counterparts. Our analysis revealed a significant divergence in the headwise entropy distributions of the LN-free GELU model (see Figure 5). While baseline models with GELU and ReLU exhibit a balanced entropy distribution, by avoiding the extreme values, the LN-free GELU model shows entropic overload in early layers.

Specifically, 58% of heads in the LN-free GELU model have entropy values between $\frac{3\max}{4}$ and max, compared to only 23% in the LN-free ReLU model (Figure 3a). More importantly, very few heads in the latter approach maximum entropy compared to the former (see yellow regions in Figure 5c), indicating more severe entropic overload in the LN-free model with GELU.

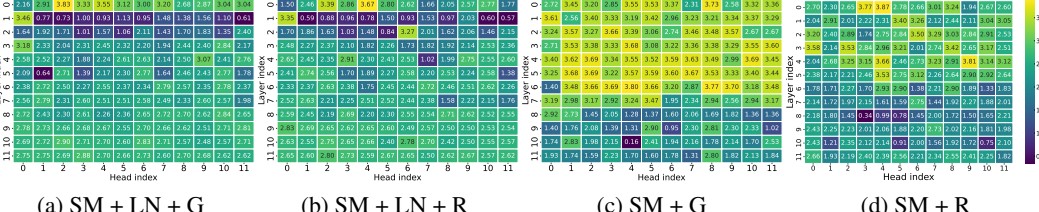

(a) SM + LN + G      (b) SM + LN + R      (c) SM + G      (d) SM + R

Figure 5: Entropy heatmaps of attention for baseline GPT-2 models with GELU and ReLU in the FFN (a and b), compared to their LayerNorm-free counterparts (c and d). In the absence of LayerNorm, using GELU in the FFN results in significantly higher entropic overload than using ReLU.

These observations align with the geometrical properties of ReLUs: they preserve more information about the structure of the raw input, encouraging neurons to specialize in different regions of the input space, leading to a higher intra-class *selectivity* and *specialization* (Alleman et al., 2024). Thus, the lack of LayerNorm makes the geometry and specialization effects of ReLU more beneficial.

## 3.2 APPROACHES TO PREVENT TRAINING COLLAPSE IN SOFTMAX-ONLY LLMS

Now, we eliminate the ReLU layer in FFN of LN-free design, resulting in a Softmax-only architecture where FFN is fully linear, and the softmax operation becomes the only source of nonlinearity in the model. We outline the key challenges in training this model and explore their potential solutions.

**Observation 3: The softmax-only model exhibits severe entropic overload in the early layers and entropy collapse in the deeper layers.** When we train the softmax-only model from scratch, the loss values quickly reach NaN and training collapses. Analyzing the layer-by-layer activation values

reveals that activations of the last few layers reach NaN very early in the training phase (Figure 6a). Further investigation into headwise entropy distribution shows that the early layers experience severe entropic overload (Figure 6b), as most of the heads in these layers are stuck at maximum entropy levels (the yellow regions). Conversely, the deeper layers suffer from entropy collapse, characterized by very low entropy values (the blue regions).

Quantitatively, 45% of total heads have entropy values in the range of $\frac{3\max}{4}$ to max, with most close to the maximum value (Figure 3a), indicating severe entropic overload. Whereas, 33% of heads exhibit values in the entropy range of 0 to $\frac{\max}{4}$, with most close to zero, indicating entropy collapse, a known indicator of training instability in transformer-based models (Zhai et al., 2023; He et al., 2024).

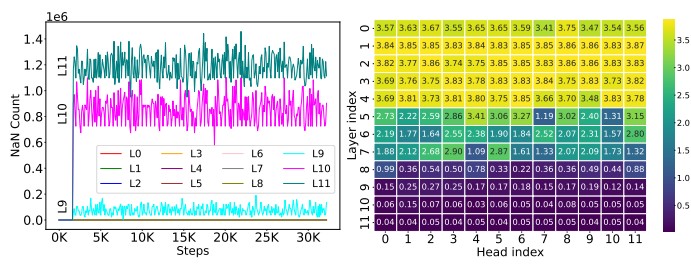

(a) Layerwise NaNs      (b) Entropy heatmap

Figure 6: Training collapses in softmax-only GPT-2 model.

**Observation 4: Normalizing the weights in FFN linear layers or appropriately scaling FFN outputs effectively prevents training collapse in softmax-only models.** To prevent training collapse while maintaining PI efficiency, we shift from activation normalization to weight normalization techniques that avoid nonlinear computations at inference. While LayerNorm requires expensive inverse-square-root operations during inference, weight normalization (Salimans & Kingma, 2016) and spectral normalization (Miyato et al., 2018) offer static alternatives. These normalization methods, normalize the weights rather than the activations, incurring no additional cost at inference.

Weight normalization reparameterizes the weight vectors as $\mathbf{W}_{\text{normalized}} = \frac{\mathbf{V}}{\|\mathbf{V}\|_2} g$, where $\mathbf{V}$ is reparameterized weight vector, $\|\mathbf{V}\|_2$ is Euclidean norm and $g$ is a learnable scaling factor. Whereas, spectral normalization normalizes the weight matrix $\mathbf{W}$ by its largest singular value $\sigma(\mathbf{W})$, yielding $\mathbf{W}_{\text{normalized}} = \frac{\mathbf{W}}{\sigma(\mathbf{W})}$. The former uses the Euclidean norm to control the magnitude of the weights during the training while the latter uses the largest singular value to constrain the Lipschitz constant of the linear layers. We employed these normalizations in the FFN of the softmax-only model which transform $\text{FFN}^{\text{SM}}(\mathbf{X}) = (\mathbf{X}\mathbf{W}_{\text{in}}^{\text{ffn}})\mathbf{W}_{\text{out}}^{\text{ffn}}$ as follows:

$$\text{FFN}_{\text{WNorm}}^{\text{SM}}(\mathbf{X}) = \left(\mathbf{X}\frac{\mathbf{V}_{\text{in}}^{\text{ffn}}}{\|\mathbf{V}_{\text{in}}\|_2}g_{\text{in}}\right)\frac{\mathbf{V}_{\text{out}}^{\text{ffn}}}{\|\mathbf{V}_{\text{out}}\|_2}g_{\text{out}} \text{ and } \text{FFN}_{\text{SNorm}}^{\text{SM}}(\mathbf{X}) = \left(\mathbf{X}\frac{\mathbf{W}_{\text{in}}^{\text{ffn}}}{\sigma(\mathbf{W}_{\text{in}}^{\text{ffn}})}\right)\frac{\mathbf{W}_{\text{out}}^{\text{ffn}}}{\sigma(\mathbf{W}_{\text{out}}^{\text{ffn}})} \quad (5)$$

Furthermore, we employ a simpler technique to scale the outputs of the FFN sub-block by having learnable scaling factors for the FFN output and their residual output as follows (see Eq. 1):

$$\mathbf{X}_{\text{out}} = \beta\hat{\mathbf{X}}_{\text{SA}} + \frac{1}{\alpha}(\text{FFN}^{\text{SM}}(\mathbf{X}_{\text{SA}})) \quad \text{where} \quad \alpha, \beta \in \mathbb{R}^L \quad (6)$$

Figure 7 demonstrates the effectiveness of these normalization techniques in stabilizing the training of softmax-only GPT-2 models by preventing entropy collapse in deeper layers. When comparing performance, we find that weight and spectral normalization led to similar performance while the learnable scaling method outperformed them with a lower perplexity (Table 3).

| | WNorm | SNorm | Scaled |
|---|---|---|---|
| Eval PPL | 3.640 | 3.624 | 3.478 |

Table 3: Perplexity comparison of weight normalization, spectral normalization, and learnable scaling employed in FFN of softmax-only GPT-2 model.

Note that the efficacy of weight or spectral normalization hinges on selecting the appropriate linear layers, as applying them to the linear layers in attention sub-block *diminishes* overall performance (see Table 9). Refer to Appendix D.1 to understand the effectiveness of the learnable scaling method.

## 4 AERO

We propose an AERO framework that tailors the existing LLM architecture by removing nonlinearity and reducing FLOPs count through targeted architectural refinements. Further, we introduce our entropy regularization technique to improve the performance of the Softmax-only model.

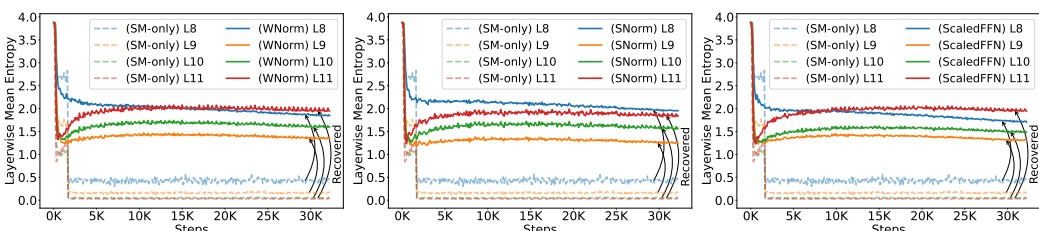

(a) Weight normalization in FFN  (b) Spectral normalization in FFN  (c) Learnable scaling of FFN outputs

Figure 7: Mitigating entropy collapse in the deeper layers of a softmax-only GPT-2 model by employing weight or spectral normalization in FFN, or by appropriately scaling FFN block outputs.

### 4.1 DESIGNING SOFTMAX-ONLY ARCHITECTURE

To eliminate nonlinearities in existing LLM architectures, we first remove normalization layers, creating an LN-free design. Our approach extends previous work on LN-free design (He et al., 2023; Noci et al., 2023; He & Hofmann, 2024) by also carefully selecting FFN activation functions, opting for ReLU due to its superior PI efficiency and ability to mitigate entropic overload in LN-free models.

We then remove ReLU, leading to a full normalization and activation-free, or Softmax-only, architecture. Training this architecture, however, poses challenges, such as entropy collapse in deeper layers. To address this, we introduce learnable scaling factors, $\alpha$ and $\beta$, in the FFN sub-block, which stabilize training more effectively than weight or spectral normalization.

### 4.2 FLOPs REDUCTION IN SOFTMAX-ONLY ARCHITECTURE

To develop an effective FLOPs reduction strategy, we begin by analyzing the distribution of FLOPs between the attention and FFN sub-blocks across varying context lengths.

**FFN FLOPs dominates in shorter context length regimes** ($T < \frac{8}{3}d$)**.** While prior work on LN-free architectures (He & Hofmann, 2024) has emphasized reducing attention FLOPs, we find that the network's FLOPs are dominated by FFN FLOPs during inference with shorter context lengths (when $T < \frac{8}{3}d$, Eq. 13). For instance, when $T \leq$1K, FFN FLOPs constitute 60%-65% of the total FLOPs in models like GPT-2 (Figure 20) and Pythia (Figure 21) variants.

Given that current research on 2PC PI primarily focuses on smaller context lengths (Zhang et al., 2025; Lu et al., 2025; Zimerman et al., 2024; Pang et al., 2024; Gupta et al., 2024; Hou et al., 2023), we strategically target reducing FFN FLOPs. First, we merge the two linear layers in FFN of Softmax-only architecture—$\mathbf{W}_{\text{in}}^{\text{ffn}} \in \mathbb{R}^{d \times 4d}$ and $\mathbf{W}_{\text{out}}^{\text{ffn}} \in \mathbb{R}^{4d \times d}$—into a single linear layer, $\mathbf{W}^{\text{ffn}} \in \mathbb{R}^{d \times d}$, as they effectively perform linear transformation in the absence of intervening nonlinearity. This reduces FFN FLOPs by a $\mathbf{8}\times$ without any performance degradation, which is not achievable in polynomial transformers, where GELU is approximated by polynomials (Zimerman et al., 2024; Li et al., 2023a).

To reduce FFN FLOPs even further, we ask the following questions: What functional role do FFNs serve when they are *purely linear*? Do all FFNs contribute *equally*, or can some of them be pruned?

**Early FFNs in Softmax-only architecture are critical, while deeper ones can be pruned.** We observe that early FFNs, despite being purely linear, are crucial for *training stability*, as their removal leads to entropy collapses (Fig. 16 and Fig. 17). Deeper FFNs, however, exhibit redundancy, allowing additional FLOPs reduction without degrading performance. This observation resonates with findings on the significance of early-to-mid (conventional non-linear) FFNs (Nanda et al., 2023; Sharma et al., 2024; Jin et al., 2024; Hu et al., 2024a; Stolfo et al., 2023; Wang et al., 2023; Haviv et al., 2023; Meng et al., 2022) and the redundant FFN computations (Kobayashi et al., 2024; Pires et al., 2023).

This enables an additional opportunity to reduce FFN FLOPs by selectively removing deeper FFNs. In Softmax-only GPT-2-small architecture, we successfully remove up to six deeper FFNs, achieving an additional $\mathbf{6}\times$ FLOPs reduction in FFN. We refer to this simplified model as `SM + ScFuFFNi`$_x$, where $x$ represents the number of deeper FFNs that are replaced with identity functions, while the remaining FFNs have one (fused) linear layer. When $x$=0, we represent the model as `SM + ScFuFFN`.

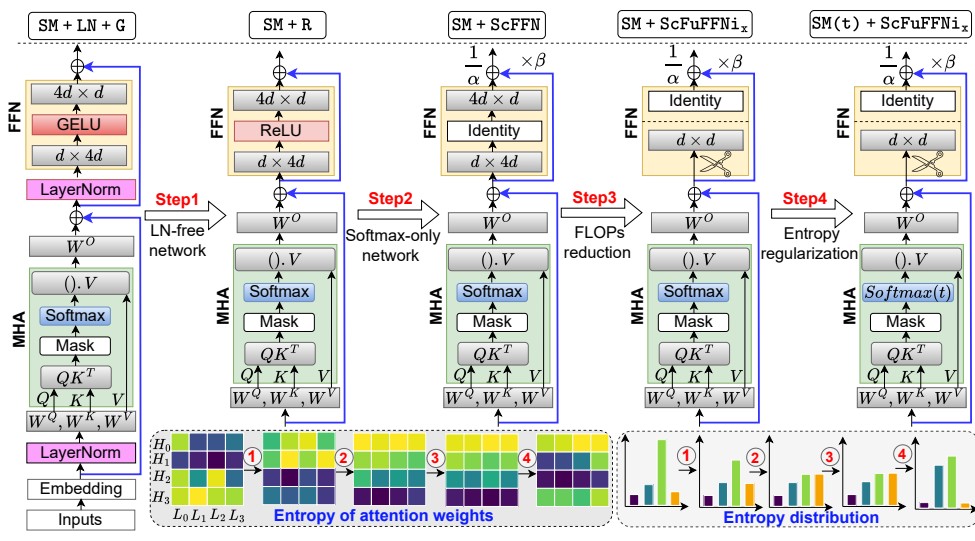

Figure 8: Overview of the proposed AERO method for reducing nonlinearities and FLOPs in transformer-based LLMs for efficient PI. The bottom of the figure shows the evolution of entropy in the attention mechanism and its distribution across attention heads.

## 4.3 ENTROPY REGULARIZATION

**Challenges in designing entropy regularization schemes to prevent entropic overload.** Previous entropy regularization approaches have primarily aimed at penalizing low-entropy predictions (Setlur et al., 2022; Pereyra et al., 2017), based on the principle of maximum entropy (Jaynes, 1982). Recently, (He et al., 2024) introduced entropy regularization to prevent entropy collapses, by addressing extremely low entropy values, in LLMs.

However, our goal is to regularize higher entropy values, which presents two-fold challenges: (1) *Head specialization:* Since each attention head captures different aspects of the input, the regularization strength needs to be adjusted for each head individually. (2) *Preventing over-regularization:* Some heads naturally exhibit higher entropy even in well-behaved entropy distributions, thus, penalizing all high-entropy values without distinction could be harmful, requiring a more flexible approach.

**Key design principles for entropy regularization.** Followings are the key design principles for our entropy regularization scheme (see Algorithm 1), addressing the aforementioned challenges:

- *Balanced entropy distribution with parameterized attention matrix:* Inspired by Miller et al. (1996), which used temperature parameter as a Lagrangian multiplier to control the entropy of a stochastic system, we parameterized the attention matrix by a learnable temperature $t \in \mathbb{R}^{H \times T}$ for each softmax operation, allowing the model to adjust the sharpness of the attention scores (see Appendix A.3). A higher temperature value ($t > 1$) diffuses the attention scores and increases the entropy, while a lower temperature value ($t < 1$) sharpens the attention scores and reduces the entropy.

- *Dynamic thresholds with head-specific adaptation:* To adapt the regularization strength based on the characteristics of each attention head (Voita et al., 2019), we use headwise learnable threshold parameter `reg_threshold_weights` $\in \mathbb{R}^H$. Consequently, the threshold for each head is computed as a learnable fraction of the maximum value of entropy (`reg_threshold_weights` $\times$ $E_{max}$), providing the fine-grained control (see Algorithm 1, line #11).

- *Tolerance margin to prevent over-regularization:* To prevent over-regularization, we allow small deviations from the respective thresholds. Thus, a penalty is imposed only if the deviation from the threshold exceeds the tolerance margin, which is set as a fraction of $E_{max}$ using the hyper-parameter $\gamma$ (see Algorithm 1, line #3).

$$\text{penalty}^{(l,h)} = \begin{cases} \left(\text{deviation}^{(l,h)}\right)^2 & \text{if } \left|\text{deviation}^{(l,h)}\right| > \gamma E_{max} \\ 0 & \text{otherwise} \end{cases}$$

The deviation from threshold is computed as $\text{deviation}^{(l,h)} = E^{(l,h)}(t) - \theta^{(l,h)} E_{max}$, where $\theta^{(l,h)}$ is `reg_threshold_weights`. The hyper-parameter $\gamma$ ensures that the model is not excessively

penalized for minor deviations from the desired entropy threshold, which could impede its capacity to learn effectively. This careful calibration between stringent regularization and desired flexibility improves the model's robustness while maintaining its adaptability to various input distributions.

- *Maximum entropy reference:* We set $E_{max} = \log(T)$ as a reference point for computing thresholds and tolerance margins to ensure consistency across different layers and heads for regularization. Additionally, it enhances interpretability by providing a quantifiable reference for measuring deviations in entropy, making the regularization process more understandable.

### 4.4 PUTTING IT ALL TOGETHER

We developed the AERO framework (Figure 8) to systematically eliminate non-linearities and reduce FFN FLOPs from the existing transformer-based LLMs. Given an input baseline LLM, the first two steps, Step1 and Step2, attempt to address the overheads associated with non-linear operations in PI, resulting in a softmax-only architecture. The next step, Step3, aims at reducing FFN FLOPs by fusing the adjacent linear layers, and then selectively pruning deeper FFNs by replacing them with identity functions, resulting in a substantial reduction in FLOPs without destabilizing the model.

Further, to mitigate the entropic overload, and improve the utilization of attention heads, Step4 introduces entropy regularization, keeping the balanced attention distributions by penalizing extreme entropy values. This step plays a crucial role in boosting the performance of the softmax-only model.

### 5 RESULTS

We conducted experiments with GPT-2 (12 and 18 layers) and Pythia-70M models on the CodeParrot and Languini book datasets, which are standard benchmarks for LLMs (He & Hofmann, 2024; He et al., 2024). For experimental setup and cryptographic protocols details, refer to Appendix C.

**Entropy regularization prevents entropic overload in Softmax-only models**

While both weight and spectral normalization, and scaling methods effectively prevent entropy collapse in the deeper layers and stabilize the training of Softmax-only models, they *fail to address the issue of entropic overload*, (see Figure 9). In contrast, the entropy regularization scheme penalizes the model to avoid extreme entropy values during training, resulting in a more balanced distribution. As a result, it complements the training stabilizing methods by further mitigating entropic overload in the early layers (see Figure 14), improving the utilization of attention heads and leading to improved performance, as demonstrated by lower perplexity.

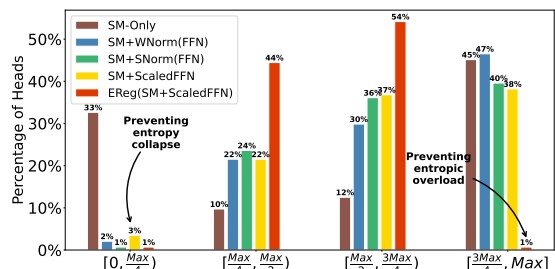

Figure 9: While normalizing weights or scaling outputs in the FFN of Softmax-only (GPT-2) model prevents entropy collapse, our proposed entropy regularization effectively mitigates entropic overload.

**Comparison of AERO vs SOTA.** We apply AERO to GPT-2, with results for each step shown in Figure 1 and a detailed analysis in Table 4. Our approach achieves up to a $4\times$ reduction in communication overhead and a $1.94\times$ speedup in end-to-end PI latency.

We also applied AERO optimizations to the LayerNorm-free design proposed in (He & Hofmann, 2024), referred to as SOTA, as they preserve model performance in their normalization-free architecture. While SOTA saves additional attention FLOPs, by introducing one extra LayerNorm layer, compared to AERO, it offers a slight speedup at the cost of significantly worse model performance, as indicated by higher perplexity. Similar observations hold for the Pythia-70M model (see Figure 15).

In terms of scalability, AERO efficiently scales to deeper models (see Table 6) and larger context lengths (see Table 5 and Table 7), whereas SOTA often suffers from training instability under these conditions. Since the contribution of MHA to the model's pre-training performance becomes more critical in the absence of FFN operations (Lu et al., 2024), we suspect that the aggressive optimization of attention FLOPs in SOTA, unlike AERO, results in inferior performance and training instability.

Table 4: Results, and comparison against SOTA (He & Hofmann, 2024), when GPT-2 ($L$=12, $H$=12, $d$=768) model is trained from scratch on CodeParrot (Face) dataset with context length 128.

| | Network Arch. | PPL | #Nonlinear Ops | #FLOPs FFN | #FLOPs Attn. | Comm. (GB) | Lat. (min.) | Savings Comm. | Savings Lat. |
|---|---|---|---|---|---|---|---|---|---|
| **Baseline** | SM + LN + G | 2.69 | SM:$144 \times \mathbb{R}^{128\times128}$ LN:$24 \times \mathbb{R}^{128\times768}$ G:$12 \times \mathbb{R}^{128\times3072}$ | 14.5B | 7.7B | 25.32 | 8.21 | 1× | 1× |
| | SM + LN + R | 2.76 | SM:$144 \times \mathbb{R}^{128\times128}$ LN:$24 \times \mathbb{R}^{128\times768}$ R:$12 \times \mathbb{R}^{128\times3072}$ | 14.5B | 7.7B | 9.44 | 6.06 | 2.68× | 1.35× |
| **SOTA** | SM + ScFFN | 4.00 | SM:$144 \times \mathbb{R}^{128\times128}$ LN: $1 \times \mathbb{R}^{128\times768}$ | 14.5B | 3.9B | 6.83 | 5.31 | 3.71× | 1.55× |
| | SM + ScFuFFN | 3.97 | SM:$144 \times \mathbb{R}^{128\times128}$ LN: $1 \times \mathbb{R}^{128\times768}$ | 1.8B | 3.9B | 6.31 | 4.50 | 4.00× | 1.82× |
| | SM + ScFuFFNi$_1$ | 4.00 | SM:$144 \times \mathbb{R}^{128\times128}$ LN: $1 \times \mathbb{R}^{128\times768}$ | 1.2B | 3.9B | 6.30 | 4.44 | 4.00× | 1.85× |
| **AERO** | SM + ScFFN | 3.50 | SM:$144 \times \mathbb{R}^{128\times128}$ | 14.5B | 7.7B | 6.95 | 5.68 | 3.64× | 1.45× |
| | SM + ScFuFFN | 3.48 | SM:$144 \times \mathbb{R}^{128\times128}$ | 1.8B | 7.7B | 6.43 | 4.76 | 3.94× | 1.72× |
| | SM + ScFuFFNi$_6$ | 3.54 | SM:$144 \times \mathbb{R}^{128\times128}$ | 0.9B | 7.7B | 6.29 | 4.23 | 4.00× | 1.94× |
| | EReg(SM(t) + ScFuFFN) | 3.21 | SM:$144 \times \mathbb{R}^{128\times128}$ | 1.8B | 7.7B | 6.43 | 4.76 | 3.94× | 1.72× |
| | EReg(SM(t) + ScFuFFNi$_6$) | 3.25 | SM:$144 \times \mathbb{R}^{128\times128}$ | 0.9B | 7.7B | 6.29 | 4.23 | 4.00× | 1.94× |

**Significance of learnable thresholds in entropy regularization** Figure 10 depicts the learnable threshold parameters (`reg_threshold_weights`) applied in the entropy regularization scheme after the model has been fully trained from scratch. They exhibit significant variability, both across layers and within individual heads of each layers, which reflects the model's ability to dynamically adjust the regularization strength in response to the specific roles of different attention heads. Such flexibility is essential for tailoring the regularization process to the distinct requirements of each head.

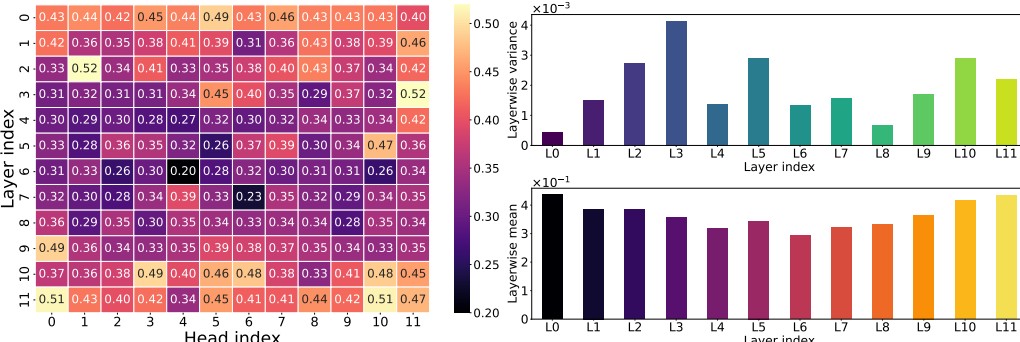

(a) Values of learned threshold weights       (b) Layerwise mean and variance of threshold weights

Figure 10: Analysis of learned threshold weights (`reg_threshold_weights`, see Eq. 4.3) in entropy regularization for softmax-only GPT-2 model: (a) Attention heads adaptively learn non-uniform threshold weights across different heads, setting individualized thresholds for entropy regularization; (b) The non-uniform means and non-zero variances across layers highlight the necessity and effectiveness of headwise learnable thresholds in adapting regularization strength.

# 6 CONCLUSION

In this work, we introduce AERO, a four-stage design framework to streamline the existing LLM architecture for efficient private inference. We design Softmax-only architecture with significantly lower FLOPs and propose entropy regularization to boost their performance.

**Limitations.** This study mainly focuses on pre-training performance, with perplexity as the primary metric, and does not include experiments to evaluate other capabilities such as transfer learning or few-shot learning. Additionally, the efficacy of the proposed Softmax-only models has been validated on models with fewer than 1B parameters. Future work will explore broader experimental evaluations, including the adaption of AERO for large-scale models (see Appendix H).

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

# Appendix

## Table of Contents

## A SHANNON ENTROPY AND ITS APPLICATION IN TRANSFORMER LLMs

### A.1 WHY USE ENTROPY TO EVALUATE THE IMPACT OF NONLINEARITIES?

We use entropy as a metric to study the impact of nonlinearities on the transformer-based LLMs for the following reasons:

- *Quantifying attention distribution:* The attention mechanism lies at the core of transformer architectures, and by computing the entropy of attention score distributions, we can observe how nonlinearities influence the spread (or the concentration) of attention scores. Higher entropy implies a more exploratory behavior, while lower entropy suggests a more focused attention distribution.
- *Feature selection:* Nonlinearities, such as ReLU, enhance feature selectivity by amplifying important features and suppressing less relevant ones (Alleman et al., 2024; Maas et al., 2013). Entropy can measure this selectivity across layers and heads, providing insights into the model's prioritization of information. Previous studies have used entropy to capture layer-wise information flow in neural networks (Peer et al., 2022).
- *Exploration vs. exploitation*: Nonlinear operators like the self-attention mechanism, LayerNorm, and GELU balance exploration and exploitation by selecting relevant features while considering a broader context. For instance, heads in the first layer focus on exploration, while those in the second layer focus on exploitation. (see Figures 5a, 5b, 12a and 12b).
- *Systematic assessment:* Prior work Zhang et al. (2024); Nahshan et al. (2024); Zhai et al. (2023); Vig & Belinkov (2019); Ghader & Monz (2017) also used entropy to analyze the behavior of transformer-based models; thus, enhancing validity and comparability of our findings.

### A.2 EVALUATING THE SHARPNESS OF ATTENTION SCORE DISTRIBUTIONS USING ENTROPY

Shannon's entropy quantifies the uncertainty in a probability distribution, measuring the amount of information needed to describe the state of a stochastic system (Shannon, 1948; Jaynes, 1957). For a probability distribution $P(x)$, the entropy is defined as $\mathbf{E}(P) = -\sum_i P(x_i) \log P(x_i)$. Refer to (Baez, 2024) for details on entropy.

In a softmax-based attention mechanism, each softmax operation yields an entropy value representing the sharpness or spread of the attention scores for each query position (Ghader & Monz, 2017; Vig & Belinkov, 2019). Higher entropy indicates a more uniform distribution of softmax scores, while lower entropy signifies a more focused distribution on certain features (Nahshan et al., 2024).

Let $\mathbf{A}^{(h,l)} \in \mathbb{R}^{T \times T}$ be the attention matrix of $h$-th head in $l$-th layer, and each element in the attention matrix, $a_{ij}^{(l,h)}$, are attention weights for the $i$-th query and $j$-th key, which are non-negative and sum to one for a query:

$$\mathbf{A}^{(l,h)} = \left[ a_{ij}^{(l,h)} \right]_{T \times T}, \quad \text{where} \quad a_{ij}^{(l,h)} \geq 0 \quad \text{and} \quad \sum_{j=1}^{T} a_{ij}^{(l,h)} = 1 \tag{7}$$

This square matrix is generated by applying the softmax operation over the key length for each query position as follows (i.e., $\mathbf{X} \in \mathbb{R}^{T \times T}$ $\mathbf{X}_i \in \mathbb{R}^{1 \times T}$):

$$\mathbf{A}^{(h,l)}(\mathbf{X}) = \text{Softmax}\left( \frac{1}{\sqrt{d_k}} (\mathbf{X}\mathbf{W}^Q)(\mathbf{X}\mathbf{W}^K)^\top \right), \quad \text{where} \quad \text{Softmax}(\mathbf{X}_i) = \frac{\exp(x_i)}{\sum_{j=1}^{T} \exp(x_j)} \tag{8}$$

Thus, each element $a_{ij}^{(l,h)}$ of the attention matrix can be represented as follows:

$$a_{ij}^{(l,h)} = \frac{\exp\left( \frac{1}{\sqrt{d_k}} (\mathbf{X}_i\mathbf{W}^Q)(\mathbf{X}_j\mathbf{W}^K)^\top \right)}{\sum_{k=1}^{T} \exp\left( \frac{1}{\sqrt{d_k}} (\mathbf{X}_i\mathbf{W}^Q)(\mathbf{X}_k\mathbf{W}^K)^\top \right)}. \tag{9}$$

Following (Zhai et al., 2023), we compute the mean of entropy values across all query positions to obtain a single entropy value for each head. The entropy $\mathbf{E}^{(l,h)}$ for the $h$-th head in the $l$-th layer of an attention matrix is given by:

$$\mathbf{E}^{(l,h)} = -\frac{1}{T} \sum_{i=1}^{T} \sum_{j=1}^{T} a_{ij}^{(l,h)} \log\left( a_{ij}^{(l,h)} + \epsilon \right) \tag{10}$$

where $\epsilon$ is a small constant added for numerical stability to prevent taking the log of zero.

### A.3 RELATIONSHIP BETWEEN TEMPERATURE AND SHANNON ENTROPY

With the learnable temperature parameters $(t)$, the attention matrix can be expressed as follows:

$$\mathbf{A}^{(l,h)}(t) = \left[ a_{ij}^{(l,h)}(t) \right]_{T \times T}, \text{ where } a_{ij}^{(l,h)}(t) = \frac{\exp\left( \frac{1}{t_i \sqrt{d_k}} (\mathbf{X}_i \mathbf{W}^Q)(\mathbf{X}_j \mathbf{W}^K)^\top \right)}{\sum_{k=1}^{T} \exp\left( \frac{1}{t_i \sqrt{d_k}} (\mathbf{X}_i \mathbf{W}^Q)(\mathbf{X}_k \mathbf{W}^K)^\top \right)}. \quad (11)$$

Let $z_{ij} = \left( \mathbf{X}_i \mathbf{W}^Q \right) \left( \mathbf{X}_j \mathbf{W}^K \right)^\top$ represents the logits (attention scores before applying softmax).

Now, substituting $a_{ij}^{(l,h)}(t)$ into the entropy formula:

$$\mathbf{E}^{(l,h)}(t) = -\frac{1}{T} \sum_{i=1}^{T} \sum_{j=1}^{T} \frac{\exp\left( \frac{1}{t\sqrt{d_k}} z_{ij} \right)}{\sum_{k=1}^{T} \exp\left( \frac{1}{t\sqrt{d_k}} z_{ik} \right)} \log\left( \frac{\exp\left( \frac{1}{t\sqrt{d_k}} z_{ij} \right)}{\sum_{k=1}^{T} \exp\left( \frac{1}{t\sqrt{d_k}} z_{ik} \right)} \right).$$

Simplifying the logarithmic term:

$$\log\left( \frac{\exp\left( \frac{1}{t\sqrt{d_k}} z_{ij} \right)}{\sum_{k=1}^{T} \exp\left( \frac{1}{t\sqrt{d_k}} z_{ik} \right)} \right) = \frac{1}{t\sqrt{d_k}} z_{ij} - \log\left( \sum_{k=1}^{T} \exp\left( \frac{1}{t\sqrt{d_k}} z_{ik} \right) \right).$$

Thus, the entropy simplifies to:

$$\mathbf{E}^{(l,h)}(t) = \frac{1}{T} \sum_{i=1}^{T} \left( \log\left( \sum_{k=1}^{T} \exp\left( \frac{1}{t\sqrt{d_k}} z_{ik} \right) \right) - \frac{1}{t\sqrt{d_k}} \sum_{j=1}^{T} a_{ij}^{(l,h)}(t) z_{ij} \right).$$

Further, it can be simplified as a function of expected value of $z_{ij}$ under the attention distribution:

$$\mathbf{E}^{(l,h)}(t) = \frac{1}{T} \sum_{i=1}^{T} \left( \log\left( \sum_{k=1}^{T} \exp\left( \frac{z_{ik}}{t\sqrt{d_k}} \right) \right) - \frac{1}{t\sqrt{d_k}} \mathbb{E}_{j \sim a_{ij}^{(l,h)}(t)} \left[ z_{ij} \right] \right) \quad (12)$$

In the above expression (Eq. 12), the first term $(\log \sum)$ represents the overall *spread* of the logits when scaled by $t$, and the second term $\left( \frac{1}{t} \mathbb{E}[z_{ij}] \right)$ represents the expected value of the scaled logits under the attention distribution.

**Temperature cases when:**

1. $t > 1$: The scaling factor $\frac{1}{t}$ reduces the influence of the logits $z_{ij}$, making the softmax distribution more uniform. Consequently, the entropy *increases*.
2. $t < 1$: The scaling factor $\frac{1}{t}$ increases the influence of the logits $z_{ij}$, making the softmax distribution more peaked. Consequently, the entropy *decreases*.
3. $t \to \infty$: The logits are scaled down to zero, and the softmax becomes a uniform distribution. The entropy reaches its maximum value of $\log T$.
4. $t \to 0$: The logits dominate the softmax, and it becomes a one-hot distribution. The entropy approaches zero.

## B INTEGRATIONS OF ENTROPY REGULARIZATION IN LOSS FUNCTION

### B.1 PYTORCH IMPLEMENTATION OF ENTROPY REGULARIZATION

The PyTorch implementation below computes the entropy regularization loss for attention weights in a transformer model. This regularization ensures a balanced attention distribution, preventing it from becoming overly concentrated or too diffuse.

PyTorch Implementation 1: Entropy Regularization Loss Calculation

```python
import torch

def calculate_entropy_reg_loss(attentions, blocks, seq_len):
    """
    Calculate the entropy regularization loss.

    Parameters:
    attentions (list): A list of attention matrices from different layers.
    blocks (list): A list of transformer blocks.
    seq_len (int): The length of the sequence (context length).

    Returns:
    float: The entropy regularization loss.
    """
    entropy_reg_loss = 0
    max_entropy = torch.log(torch.tensor(seq_len)) # Theoretical maximum
        entropy
    fraction = 0.10 # Design hyper-parameter for tolerance margin
    tolerance_margin = fraction * max_entropy # Set tolerance margin as
        fraction of the maximum entropy

    for layer_idx, (block, attn_mat) in enumerate(zip(blocks, attentions)):

        reg_threshold_weights = block.attn.reg_threshold_weights # Head-
            wise learnable parameters to set head-specific threshold
        ent_val = -torch.sum(attn_mat * torch.log(attn_mat + 1e-9), dim=-1)
            # Compute entropy averaged over sequence length
        layer_entropy_reg_loss = 0

        for head_idx in range(block.attn.num_heads):
            head_entropy = ent_val[:, head_idx, :] # Get head-specific
                entropy
            threshold = reg_threshold_weights[head_idx] * max_entropy
            deviation = torch.abs(head_entropy - threshold)
            penalty = torch.square(torch.where(deviation > tolerance_margin,
                deviation, torch.zeros_like(deviation)))
            layer_entropy_reg_loss += penalty.sum()

        layer_entropy_reg_loss /= block.attn.num_heads
        entropy_reg_loss += layer_entropy_reg_loss

    entropy_reg_loss /= len(attentions)
    return entropy_reg_loss

# Calculate the total loss including entropy regularization
lambda_reg = 1e-5 # Hyperparameter for entropy regularization weight
entropy_regularization = calculate_entropy_reg_loss(attentions, blocks,
    seq_len)
total_loss = ce_loss + lambda_reg * entropy_regularization
```

## B.2 Entropy Regularization Algorithm

## C Design of Experiments

**System setup** We use a SecretFlow setup (Lu et al., 2025) with the client and server simulated on two physically separate machines, each equipped with an AMD EPYC 7502 server with specifications of 2.5 GHz, 32 cores, and 256 GB RAM. We measure the *end-to-end* PI latency, including input

---

**Algorithm 1** Entropy Regularization Loss Computation

---

**Inputs:** attentions: List of attention matrices, $\Theta(L, H)$= reg_threshold_weights, $T$: Sequence length, $\lambda$: Regularization loss weightage, $\gamma$: Hyper-parameter for Tolerance margin
**Output:** $\mathcal{L}_{\text{total}}$: Total loss including entropy regularization

1: $\mathcal{L}_{\text{entropy}} \leftarrow 0$
2: $\text{E}_{\max} \leftarrow \log(T)$             ▷ Theoretical maximum value of entropy
3: $\text{Tol}_{\text{margin}} \leftarrow \gamma \text{E}_{\max}$         ▷ Tolerance margin is set as a small fraction of $\text{E}_{\max}$
4: **for** each layer $l$ in layers **do**
5:      $\mathcal{L}_{\text{layer}} \leftarrow 0$
6:      $\text{A}(t) \leftarrow \text{attentions}[l]$   ▷ Attention matrix with learnable temperature for each query position
7:      $\text{E}(t) \leftarrow -\frac{1}{T}\sum_{i=1}^{T}\sum_{j=1}^{T} \text{A}_{ij}(t)\log(\text{A}_{ij}(t))$ ▷ Compute entropy, averaged over query length
8:      **for** each head $h$ in heads **do**
9:          $E^{(l,h)} \leftarrow \text{Slice}(\text{E}(t), h)$            ▷ Entropy for head $h$
10:         $\theta^{(l,h)} \leftarrow \text{Slice}(\Theta(L, H), h)$         ▷ Learnable threshold weight head $h$
11:         $\delta^{(l,h)} \leftarrow \text{E}^{(l,h)}(t) - \theta^{(l,h)}\text{E}_{\max}$     ▷ Deviation from head-specific threshold
12:         $\text{penalty}^{(l,h)} \leftarrow (\delta^{(l,h)})^2 \mathbb{1}(|\delta^{(l,h)}| > \text{Tol}_{\text{margin}})$ ▷ Penalize iff deviation exceeds Tolerance
13:         $\mathcal{L}_{\text{layer}} \leftarrow \mathcal{L}_{\text{layer}} + \text{penalty}^{(l,h)}$
14:      **end for**
15:      $\mathcal{L}_{\text{layer}} \leftarrow \frac{\mathcal{L}_{\text{layer}}}{\text{num\_heads}}$            ▷ Average over heads
16:      $\mathcal{L}_{\text{entropy}} \leftarrow \mathcal{L}_{\text{entropy}} + \mathcal{L}_{\text{layer}}$
17: **end for**
18: $\mathcal{L}_{\text{entropy}} \leftarrow \frac{\mathcal{L}_{\text{entropy}}}{\text{len(attentions)}}$            ▷ Average over layers
19: $\mathcal{L}_{\text{total}} \leftarrow \mathcal{L}_{\text{CE}} + \lambda \mathcal{L}_{\text{entropy}}$
20: **return** $\mathcal{L}_{\text{total}}$

---

embeddings and final output (vocabulary projection) layers, in WAN setting (bandwidth:100Mbps, latency:80ms), simulated using Linux Traffic Control (tc) commands. The number of threads is set to 32. Following He & Hofmann (2024); Stanić et al. (2023); Geiping & Goldstein (2023), all the models are trained on a single RTX 3090 GPU.

**Datasets** We train models from scratch using the CodeParrot Face and Languini book Stanić et al. (2023) datasets. The CodeParrot dataset, sourced from 20 million Python files on GitHub, contains 8 GB of files with 16.7 million examples, each with 128 tokens, totaling 2.1 billion training tokens. We use a tokenizer with a vocabulary of 50K and train with context lengths of 128 and 256. The Languini book dataset includes 84.5 GB of text from 158,577 books, totaling 23.9 billion tokens with a WikiText-trained vocabulary of 16,384, and train with context length of 512. Each book averages 559 KB of text or about 150K tokens, with a median size of 476 KB or 128K tokens.

**Training Hyperparameters** For pre-training on the CodeParrot dataset, we adopt the training settings from (He & Hofmann, 2024). Similarly, for training on the Languini dataset, we follow the settings from (Stanić et al., 2023). These settings remain consistent across all architectural variations to accurately reflect the impact of the architectural changes. When applying entropy regularization on the CodeParrot dataset, we initialize the learnable temperature to 1e-2 and set $\lambda$ to 1e-5. For the Languini dataset, the temperature is initialized to 1e-1, and $\lambda$ is set to 5e-5.

## C.1    PERPLEXITY AS A RELIABLE METRIC TO EVALUATE THE LLMS' PERFORMANCE

Perplexity (Jelinek et al., 1977) is a widely adopted metric to evaluate the predictive performance of auto-regressive language models, reflecting the model's ability to predict the next token in a sequence. However, for perplexity to serve as a meaningful comparative metric across different architectures, it is critical to ensure consistency in the tokenizer, and vocabulary size and quality (Hutchins et al., 2022). Any variation in these components can potentially skew the results by inflating or deflating perplexity scores; thus, obfuscating the true effects of architectural changes.

In our work, we maintain tokenization schemes and vocabulary attributes as invariant factors across all experiments within a dataset. This isolation of architectural modifications ensures that any observed variations in perplexity are directly attributable to changes in the model design. Thus, by enforcing a

consistent tokenization scheme and vocabulary within a dataset, we ensure that perplexity remains a reliable metric for comparing model architectures. Consequently, lower perplexity in our evaluations reliably reflects improved token-level predictions.

## C.2 WHY TRAINING FROM SCRATCH TO STUDY NONLINEARITIES?

Understanding the intricate roles of architectural components and nonlinearities—such as activation functions (e.g., GELU, ReLU) in FFN, normalization layers (e.g., LayerNorm), etc.—in transformer-based language models necessitates a methodical and detailed investigative approach. Training models from scratch is essential for this purpose, as it allows us to delve into the internal mechanisms of the model using quantitative measures like entropy. Below, we present a justification for our methodology:

- *Nonlinearities' impact on the fundamental learning dynamics:* Nonlinearities significantly influence the optimization landscape by affecting gradient flow and the model's ability to navigate non-convex loss surfaces. Training models from scratch allow us to observe the fundamental learning dynamics that emerge during the initial stages of training. Thus, constructing models with controlled variations, such as substituting or excluding specific nonlinearities, enables us to isolate their direct effects impact on convergence behavior and training stability.
- *Understanding internal mechanisms through entropy analysis:* Training from scratch enables us to navigate the evolution of entropy values across the layers and assess how architectural components influence information flow within the model. This analysis provides deep insights into the internal workings of models that may not be accessible when starting from pre-trained checkpoints.
- *Limitations of fine-tuning approaches:* The aforementioned granular level of analysis is unattainable when starting from pre-trained models, where the optimization trajectory has already been largely determined. In contrast, training models from scratch eliminates confounding variables that could arise from pre-existing weights and learned representations, ensuring that any observed effects are solely due to the architectural modifications introduced.

## C.3 CRYPTOGRAPHIC PROTOCOLS FOR LINEAR AND NONLINEAR OPERATIONS

**Linear Operations (MatMul)** For privacy-preserving matrix multiplication operations, BumbleBee leverages homomorphic encryption with a novel ciphertext compression strategy Lu et al. (2025). The protocol implements Oblivious Linear Transform (OLT) with efficient packing techniques that reduce communication costs by 80-90% compared to previous approaches. Testing on BERT-base model showed 92% less communication compared to IRON Hao et al. (2022) and 90% less than BOLT Pang et al. (2024). The protocol seamlessly handles both scenarios where one matrix is in plaintext and another is secret-shared, as well as when both matrices are secret-shared, adapting its compression strategy based on matrix dimensions.

**GELU** To efficiently compute GELU activation, the protocol employs strategic polynomial approximations (with degree 3 and degree 6) across different input ranges. The implementation optimizes branch selection through batched comparisons and leverages mixed bitwidth arithmetic, achieving 89% reduction in communication costs compared to previous approaches. This optimization maintains model accuracy within 1% of plaintext evaluation while being 35% faster in computation time compared to BOLT's implementation.

**ReLU** The private ReLU is implemented through a composition of secure comparison and multiplexer operations. Using optimized OT(oblivious transfer)-based comparison protocols and Boolean-to-arithmetic conversions, achieving significant efficiency gains. The protocol leverages Ferret OT Yang et al. (2020) instead of traditional IKNP OT Ishai et al. (2003), contributing to the overall improvement in communication efficiency while maintaining the simplicity of ReLU computation.

**LayerNorm** The LayerNorm computation breaks down into optimized sub-components: secure mean calculation, variance computation using an efficient square protocol, and normalization using reciprocal square root. The implementation takes advantage of squaring operations costing half of general multiplication in secure computation. This optimization, combined with efficient reciprocal computation, shows significant improvement over IRON's approach Hao et al. (2022), contributing to the overall $13\times$ speedup in end-to-end inference time.

**Softmax** For Softmax computation, the protocol implements a numerically stable approach using max normalization. The exponential function computation is optimized specifically for negative inputs using Taylor approximation, reducing communication by 80% compared to previous approaches. The division operation is restructured to use reciprocal followed by multiplication, resulting in over 80% reduction in communication costs while maintaining numerical precision.

For detailed protocol specifications and security proofs, refer to the BumbleBee paper Lu et al. (2025).

## D  ADDITIONAL RESULTS

### D.1  LEARNABLE SCALING FACTORS IN SCALED-FFN OF SOFTMAX-ONLY ARCHITECTURE

We plot the values of FFN scaling factors $\alpha$ and $\beta$ (see Eq. 6) learned across the layers in the full-trained softmax-only GPT-2 model, and made the following observations from the Figure 11:

- **Significant increase in $\alpha$ with layer depth:** The scaling factor $\alpha$ increases substantially in the deeper layers of the model, with particularly high values observed in L10. This indicates that as the network goes deeper, the FFN outputs are heavily scaled down by $\alpha$. This downscaling is essential to prevent the FFN outputs from dominating the activations, which could otherwise lead to numerical instability, as evidenced by the NaNs observed early in training in Figure 6a. The large $\alpha$ values in deeper layers suggest that this downscaling becomes increasingly critical as the model progresses through its layers, effectively stabilizing the training process by keeping the FFN outputs in check.
- **Balancing $\beta$ values across layers:** The $\beta$ scaling factors, which modulate the residual connections within the FFN sub-block by up-scaling their output, start higher in the earlier layers and gradually decrease, with some fluctuation, as the layers deepen. The moderate up-scaling provided by $\beta$ helps to ensure that the residual connections are not overshadowed by the scaled-down FFN outputs. This balance between the strong downscaling by $\alpha$ and the corresponding upscaling by $\beta$ is crucial for maintaining stable activations across layers, particularly in deeper layers where instability is most likely to occur.

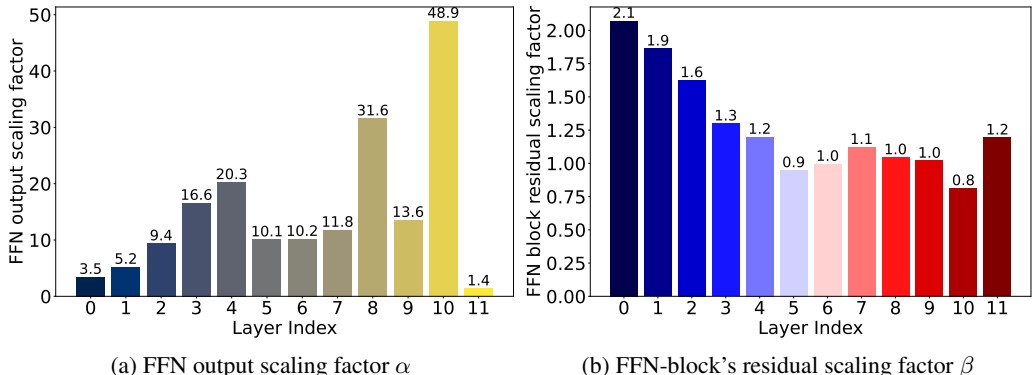

(a) FFN output scaling factor $\alpha$        (b) FFN-block's residual scaling factor $\beta$

Figure 11: Learned scaling factors $\alpha$ and $\beta$ in Eq. 6 across different layers in the Softmax-only GPT-2 model ($L$=12, $H$=12, $d$=768). The values were plotted after full model training to observe the modulation of FFN outputs and residual connections in each layer.

These observations underscore the critical role that the learnable scaling factors $\alpha$ and $\beta$ play in stabilizing the training of softmax-only GPT-2 models. By dynamically adjusting the contributions of the FFN sub-block outputs, $\alpha$ and $\beta$ prevent the numerical issues that arise in deeper layers, ensuring stable and effective training. This fine-tuned balance is key to avoiding entropy collapse and other forms of instability that would otherwise derail the training process.

## D.2 ENTROPY DYNAMICS IN LLM ARCHITECTURE WITH FEWER NONLINEARITY

Figure 12 presents the entropy dynamics of the GPT-2 model as nonlinearities are progressively removed, with the models trained from scratch. In Figure 13, the entropy dynamics are shown for a normalization-free GPT-2 model with a learnable negative slope in leaky ReLU of FFN. Figure 14 represents the entropy dynamics when various methods of mitigating the training instability (weight and spectral normalization in FFN, and learnable scaling factors for FFN outputs) in Softmax-only GPT-2 modes are applied, and also for entropy-regularization which is applied to overcome the entropic overload.

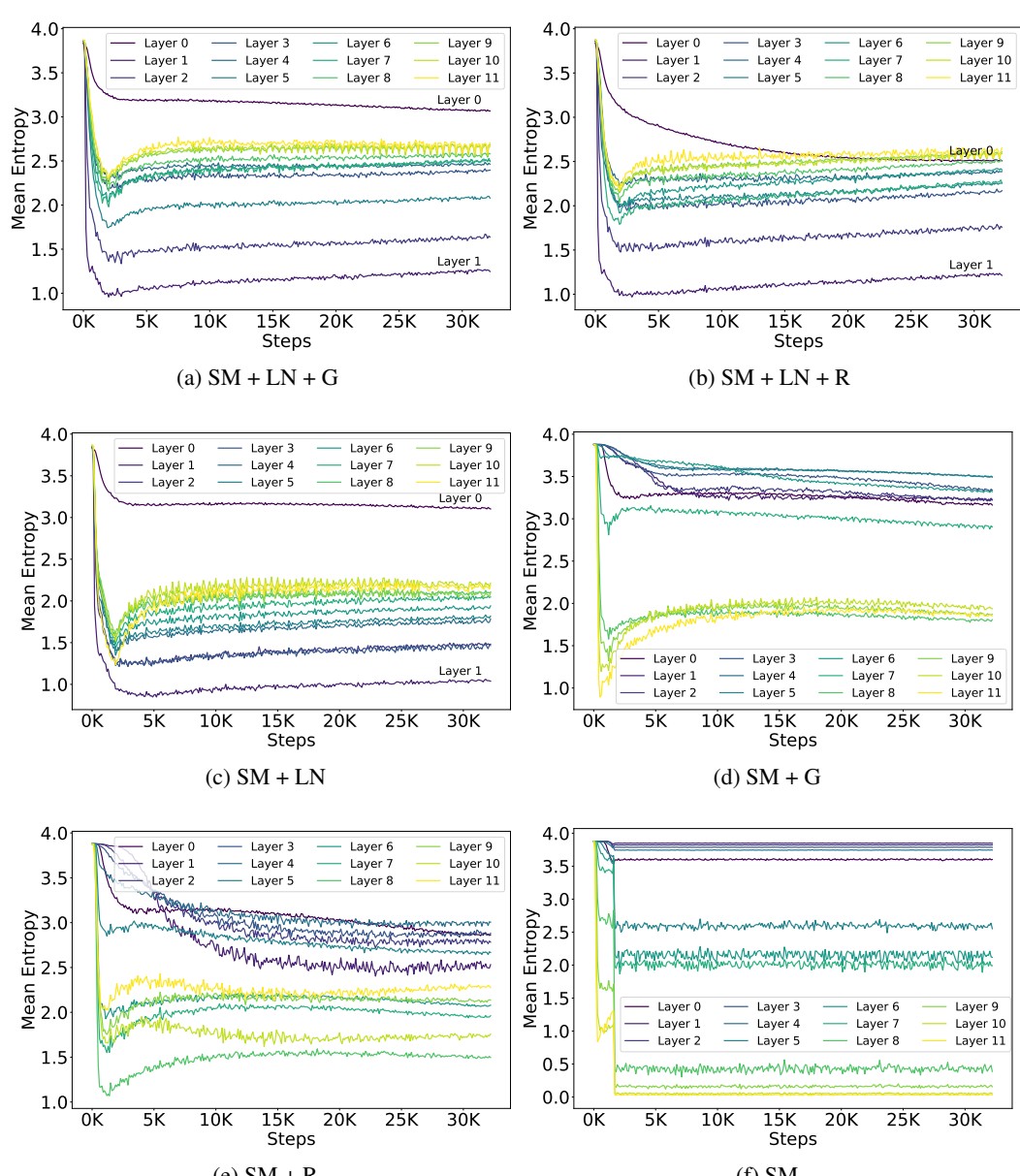

Figure 12: Evolution of Layerwise entropy when GPT-2 ($L$=12, $H$=12, $d$=768) models with various nonlinearity configurations are trained from scratch on CodeParrot dataset.

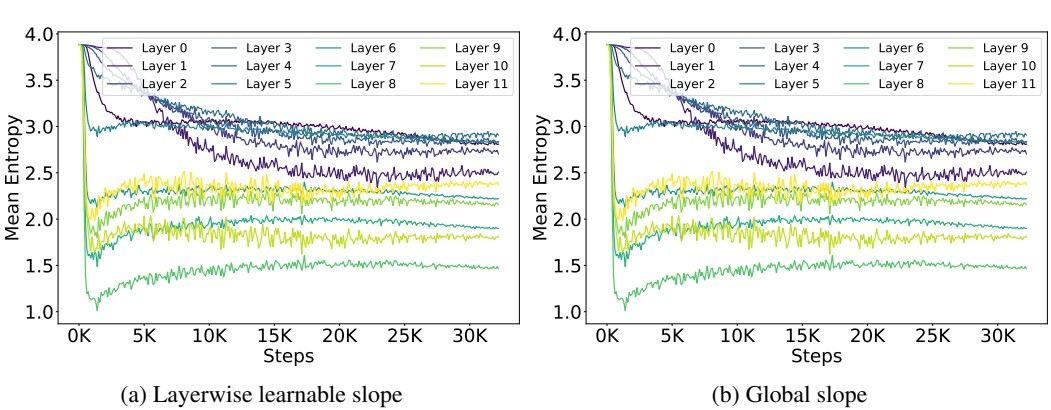

(a) Layerwise learnable slope

(b) Global slope

Figure 13: Evolution of layerwise entropy in LayerNorm-free GPT-2 models ($L = 12$, $H = 12$, $d = 768$) with a learnable negative slope in the leaky ReLU activation function, trained from scratch on the CodeParrot dataset

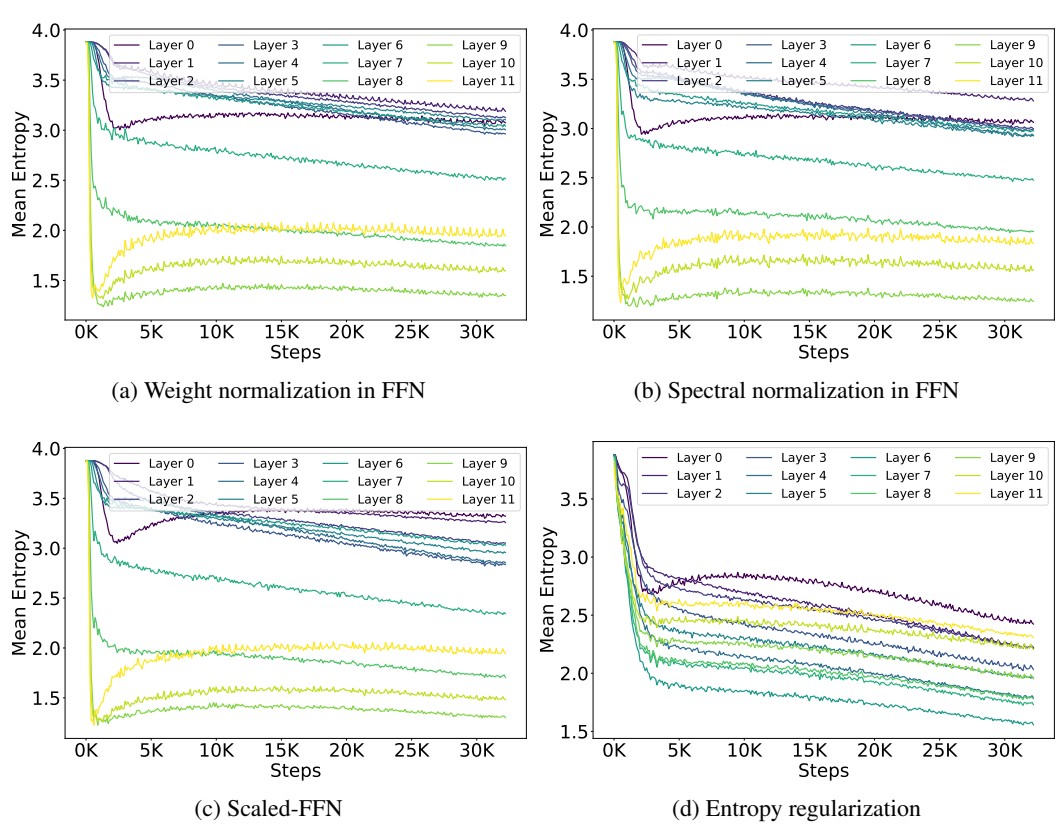

(a) Weight normalization in FFN

(b) Spectral normalization in FFN

(c) Scaled-FFN

(d) Entropy regularization

Figure 14: "Layerwise entropy evolution in Softmax-only GPT-2 models ($L = 12$, $H = 12$, $d = 768$), trained from scratch on the CodeParrot dataset. Weight and spectral normalization techniques, along with learnable scaling factors (scaled-FFN), are applied in the FFN. In the final configuration (d), entropy regularization is used within the scaled-FFN to address the entropic overload observed in early layers. Notably, the entropy in early layers of (d) is lower compared to (a), (b), and (c).

### D.3 ADDITIONAL RESULTS FOR LATENCY AND COMMUNICATION SAVINGS USING AERO

**GPT-2 Model with 256 tokens as input context** Table 5 provides an analysis of the latency and communication savings achieved by applying AERO to the GPT-2 model with 256 context length, along with a detailed breakdown of the nonlinear operations and FLOPs. The performance of AERO is also compared against SOTA.

Table 5: Results, and comparison against SOTA (He & Hofmann, 2024), when GPT-2 ($L$=12, $H$=12, $d$=768) model is trained from scratch on CodeParrot (Face) dataset with context length 256. NaNs indicate training instability in SOTA.

| | Network Arch. | PPL | #Nonlinear Ops | #FLOPs FFN | #FLOPs Attn. | Comm. (GB) | Lat. (min.) | Savings Comm. | Savings Lat. |
|---|---|---|---|---|---|---|---|---|---|
| Baseline | SM + LN + G | 2.35 | SM:$144 \times \mathbb{R}^{256 \times 256}$ LN:$24 \times \mathbb{R}^{256 \times 768}$ G:$12 \times \mathbb{R}^{256 \times 3072}$ | 29.0B | 16.3B | 58.51 | 16.57 | 1× | 1× |
| Baseline | SM + LN + R | 2.41 | SM:$144 \times \mathbb{R}^{256 \times 256}$ LN:$24 \times \mathbb{R}^{256 \times 768}$ R:$12 \times \mathbb{R}^{256 \times 3072}$ | 29.0B | 16.3B | 26.73 | 12.59 | 2.19× | 1.32× |
| SOTA | SM + ScFFN | 3.47 | SM:$144 \times \mathbb{R}^{256 \times 256}$ LN: $1 \times \mathbb{R}^{256 \times 768}$ | 29.0B | 8.5B | 21.52 | 11.42 | 2.72× | 1.45× |
| SOTA | SM + ScFuFFN | NaNs | SM:$144 \times \mathbb{R}^{256 \times 256}$ LN: $1 \times \mathbb{R}^{256 \times 768}$ | 3.6B | 8.5B | 20.48 | 10.14 | 2.86× | 1.63× |
| AERO | SM + ScFFN | 3.04 | SM:$144 \times \mathbb{R}^{256 \times 256}$ | 29.0B | 16.3B | 21.77 | 11.91 | 2.69× | 1.39× |
| AERO | SM + ScFuFFN | 3.03 | SM:$144 \times \mathbb{R}^{256 \times 256}$ | 3.6B | 16.3B | 20.72 | 10.45 | 2.82× | 1.59× |
| AERO | SM + ScFuFFNi$_6$ | 3.08 | SM:$144 \times \mathbb{R}^{256 \times 256}$ | 1.8B | 16.3B | 20.59 | 10.32 | 2.84× | 1.61× |
| AERO | EReg(SM(t) + ScFuFFN) | 2.92 | SM:$144 \times \mathbb{R}^{256 \times 256}$ | 3.6B | 16.3B | 20.72 | 10.45 | 2.82× | 1.59× |
| AERO | EReg(SM(t) + ScFuFFNi$_6$) | 2.97 | SM:$144 \times \mathbb{R}^{256 \times 256}$ | 1.8B | 16.3B | 20.59 | 10.32 | 2.84× | 1.61× |

**GPT-2 Model with 18 Layers** Table 6 provides an analysis of the latency and communication savings achieved by applying AERO to the 18-layer GPT-2 model, along with a detailed breakdown of the nonlinear operations and FLOPs. The performance of AERO is also compared against SOTA.

Table 6: Results, and comparison against SOTA (He & Hofmann, 2024), when GPT-2 ($L$=18, $H$=12, $d$=768) model is trained from scratch on CodeParrot (Face) dataset with context length 128. NaNs indicate training instability in SOTA.

| | Network Arch. | PPL | #Nonlinear Ops | #FLOPs FFN | #FLOPs Attn. | Comm. (GB) | Lat. (min.) | Savings Comm. | Savings Lat. |
|---|---|---|---|---|---|---|---|---|---|
| Baseline | SM + LN + G | 2.56 | SM:$216 \times \mathbb{R}^{128 \times 128}$ LN:$36 \times \mathbb{R}^{128 \times 768}$ G:$18 \times \mathbb{R}^{128 \times 3072}$ | 21.7B | 11.6B | 37.17 | 10.77 | 1× | 1× |
| Baseline | SM + LN + R | 2.63 | SM:$216 \times \mathbb{R}^{128 \times 128}$ LN:$36 \times \mathbb{R}^{128 \times 768}$ R:$18 \times \mathbb{R}^{128 \times 3072}$ | 21.7B | 11.6B | 13.34 | 8.04 | 2.79× | 1.34× |
| SOTA | SM + ScFFN | NaNs | SM:$216 \times \mathbb{R}^{128 \times 128}$ LN: $1 \times \mathbb{R}^{128 \times 768}$ | 21.7B | 5.9B | 9.39 | 6.75 | 3.96× | 1.60× |
| AERO | SM + ScFFN | 3.26 | SM:$216 \times \mathbb{R}^{128 \times 128}$ | 21.7B | 11.6B | 9.62 | 7.23 | 3.86× | 1.49× |
| AERO | SM + ScFuFFN | 3.24 | SM:$216 \times \mathbb{R}^{128 \times 128}$ | 2.7B | 11.6B | 8.83 | 6.07 | 4.21× | 1.77× |
| AERO | SM + ScFuFFNi$_4$ | 3.27 | SM:$216 \times \mathbb{R}^{128 \times 128}$ | 2.1B | 11.6B | 8.79 | 5.85 | 4.23× | 1.84× |
| AERO | EReg(SM(t) + ScFuFFN) | 3.13 | SM:$216 \times \mathbb{R}^{128 \times 128}$ | 2.7B | 11.6B | 8.83 | 6.07 | 4.21× | 1.77× |
| AERO | EReg(SM(t) + ScFuFFNi$_4$) | 3.17 | SM:$216 \times \mathbb{R}^{128 \times 128}$ | 2.1B | 11.6B | 8.79 | 5.85 | 4.23× | 1.84× |

### D.4 RESULTS ON LANGUINI DATASET

Table 7 provides an analysis of the latency and communication savings achieved by applying AERO to the GPT-2 model on Languini dataset, trained on 1.2B, 2.4B, and 4.8B tokens. We also provides a detailed breakdown of the nonlinear operations and FLOPs.

Table 7: Results, and comparison against SOTA (He & Hofmann, 2024), when GPT-2 ($L$=12, $H$=12, $d$=768) model is trained from scratch on Languini (Stanić et al., 2023) dataset with context length 512. NaNs indicate training instability in SOTA.

| | Network Arch. | Eval PPL | | | #Nonlinear Ops | #FLOPs | | Comm. (GB) | Lat. (min.) |
|---|---|---|---|---|---|---|---|---|---|
| | | 1.2B | 2.4B | 4.8B | | FFN | Attn. | | |
| Baseline | SM + LN + G | 25.71 | 23.32 | 21.29 | SM:$144 \times \mathbb{R}^{512 \times 512}$ LN:$24 \times \mathbb{R}^{512 \times 768}$ G:$12 \times \mathbb{R}^{512 \times 3072}$ | 58.0B | 36.2B | 145.24 | 30.74 |
| | SM + LN + R | 26.06 | 23.55 | 21.58 | SM:$144 \times \mathbb{R}^{512 \times 512}$ LN:$24 \times \mathbb{R}^{512 \times 768}$ R:$12 \times \mathbb{R}^{512 \times 3072}$ | 58.0B | 36.2B | 81.71 | 23.54 |
| SOTA | SM + ScFFN | NaNs | NaNs | NaNs | SM:$144 \times \mathbb{R}^{512 \times 512}$ LN: $1 \times \mathbb{R}^{512 \times 768}$ | 58.0B | 19.3B | 72.10 | 21.56 |
| AERO | SM + ScFFN | 33.91 | 31.12 | 28.89 | SM:$144 \times \mathbb{R}^{512 \times 512}$ | 58.0B | 36.2B | 71.76 | 21.51 |
| | SM + ScFuFFN | 33.77 | 30.82 | 28.59 | SM:$144 \times \mathbb{R}^{512 \times 512}$ | 7.3B | 36.2B | 69.68 | 19.44 |
| | SM + ScFuFFNi$_1$ | 34.16 | 31.23 | 29.69 | SM:$144 \times \mathbb{R}^{512 \times 512}$ | 6.6B | 36.2B | 69.64 | 19.11 |
| | EReg(SM(t) + ScFuFFN) | 31.54 | 28.70 | 26.55 | SM:$144 \times \mathbb{R}^{512 \times 512}$ | 7.3B | 36.2B | 69.68 | 19.44 |
| | EReg(SM(t) + ScFuFFNi$_1$) | 31.75 | 28.93 | 26.74 | SM:$144 \times \mathbb{R}^{512 \times 512}$ | 6.6B | 36.2B | 69.64 | 19.11 |

## D.5 RESULTS ON PYTHIA-70M

Table 8 presents the pre-training performance, measured in terms of perplexity, for input context lengths of 128 and 256. The results show the impact of progressively removing nonlinearities from the Pythia-70M model.

Table 8: Results for nonlinearity reduction in Pythia-70M ($L$=6, $H$=8, $d$=512) on CodeParrot dataset. LN-free architecture with ReLU activations in FFN (i.e., SM + R) significantly outperform their GELU counterpart (i.e., SM + G) across various content length ($T$).

| | $T$=128 | | $T$=256 | |
|---|---|---|---|---|
| | Eval PPL | $+\Delta(\%)$ | Eval PPL | $+\Delta(\%)$ |
| SM+LN+G | 3.512 | 0.00 | 3.054 | 0.00 |
| SM+LN+R | 3.590 | 2.22 | 3.107 | 1.73 |
| SM+LN | 4.445 | 26.56 | 3.836 | 25.60 |
| SM+G | 4.086 | 16.35 | 3.570 | 16.87 |
| SM+R | 3.736 | 6.36 | 3.273 | 7.17 |

Figure 15 illustrates the step-wise latency and communication savings achieved by applying AERO, as well as its impact on perplexity, on Pythia-70M models, also, compares with the SOTA at iso-latencies.

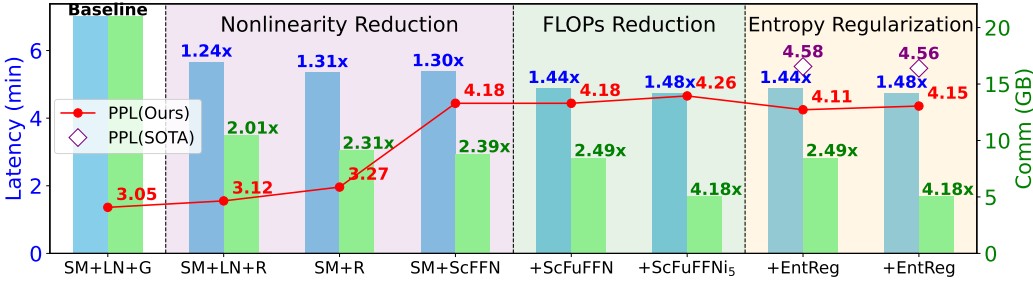

Figure 15: Latency and communication savings through nonlinearity and FLOPs reduction steps when AERO is applied on Pythia-70M, and model is trained from scratch on CodeParrot dataset with context length 256. Further, we benchmark AERO against the SOTA He & Hofmann (2024) at iso-latency points.

## D.6 PERFORMANCE COMPARISON OF WEIGHT AND SPECTRAL NORMALIZATION

Table 9 compares the performance of weight and spectral normalization applied in various linear layers within the attention and FFN sub-blocks. The results show that applying these techniques to the attention blocks yields diminishing returns compared to their application in the FFN.

Table 9: Comparison of weight normalization Salimans & Kingma (2016) and spectral normalization Miyato et al. (2018) when employed in Softmax-only GPT-2 ($L$=12, $H$=12, $d$=768) models, and trained from scratch on CodeParrot dataset with 128 input context length. FFN weight normalization yield the similar results; whereas, weight normalization works better in other linear layers.

| Linear layers | Eval PPL(WNorm) | Eval PPL(SNorm) |
|---|---|---|
| QK | 3.89 | 4.25 |
| FFN | 3.64 | 3.63 |
| QK+FFN | 3.88 | 4.23 |
| QKV+FFN | 3.93 | 4.26 |
| QKVO+FFN | 3.98 | 4.34 |

## D.7 TRAINING DYNAMICS IN SOFTMAX-ONLY MODELS WITH FEWER FFNS

To further reduce the FLOPs in the $\mathtt{SM} + \mathtt{ScFuFFN}$ model, where each FFN is simplified to a single fused linear layer, we experimented with gradually pruning the deeper FFNs by replacing them with identity connections and monitoring training stability. Figure 16 presents a comparison of headwise entropy distributions between the pruned and unpruned models, both trained from scratch. Instability emerged when more than 6 deeper FFNs were pruned, as indicated by a significant shift in the headwise entropy distribution. Specifically, we observed entropy collapse, where a disproportionate number of atten-

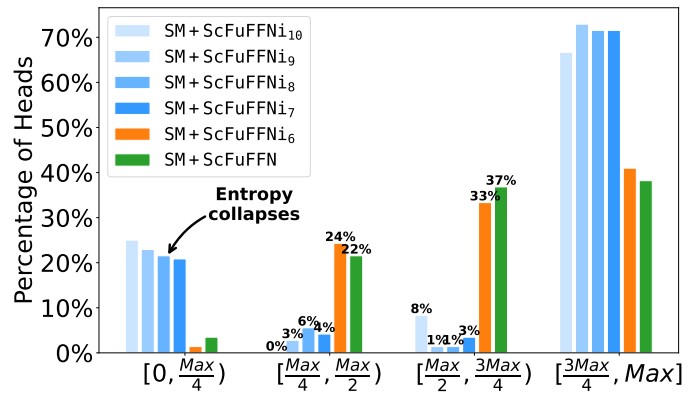

Figure 16: Head-wise entropy distribution in the Softmax-only GPT-2 model ($L$=12, $H$=12, $d$=768) with earlier FFNs intact and deeper FFNs pruned, trained from scratch on the CodeParrot dataset.

tion heads exhibited extremely low entropy values, ranging from 0 to $\frac{\mathtt{Max}}{4}$, with most values near zero. Less than 10% of the attention heads maintained entropy values in the balanced range of $\frac{\mathtt{Max}}{4}$ to $\frac{3\mathtt{Max}}{4}$. This highlights the critical role of earlier FFNs in stabilizing the training of the model, even when reduced to linear transformations.

To investigate this instability further, Figure 17 provides a detailed analysis. Training stability is maintained when up to 6 FFNs are pruned, as shown by the layer-wise entropy dynamics and heatmaps, which resembles the behavior of the unpruned $\mathtt{SM} + \mathtt{ScFuFFN}$ model. In particular, both models have approximately 55% to 60% of attention heads exhibiting entropy values in the balanced range, with negligible attention heads falling within the 0 to $\frac{\mathtt{Max}}{4}$ range. However, pruning the 7th FFN ($\mathtt{SM} + \mathtt{ScFuFFN}i_7$) leads to a sudden shift, resulting in NaNs and entropy collapse, particularly in the deeper layers. The similarity in entropy dynamics and heatmaps between the stable configurations suggests that the model remains robust as long as no more than 6 FFNs are pruned.

Nonetheless, all the pruned and unpruned models exhibit entropic overload, where a significant fraction of attention heads possess high entropy values in the range $\frac{3\mathtt{Max}}{4}$ to $\mathtt{Max}$, stable models exhibiting this overload to a lesser extent.

## D.8 MITIGATING OVER-REGULARIZATION WITH AN APPROPRIATE THRESHOLD MARGIN

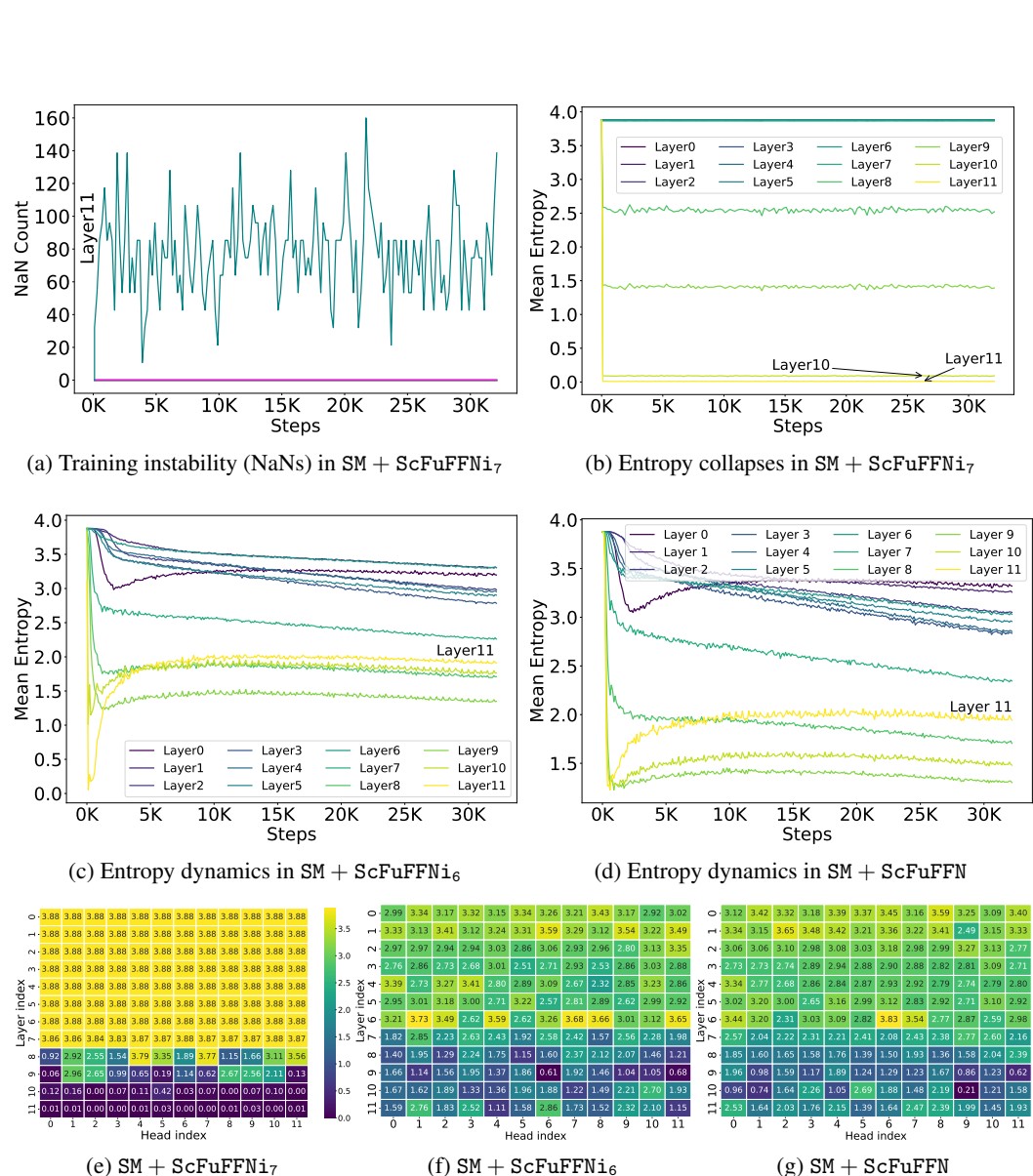

Figure 17: Training instability is evident with NaNs in the final layers (a) and entropy collapse in the last two layers (b) of the SM + ScFuFFNi₇ configuration, where 7 deeper FFNs are pruned in the Softmax-only GPT-2 model ($L = 12$, $H = 12$, $d = 768$), trained from scratch on the CodeParrot dataset. In contrast, stable training is observed in (c) with no entropy collapse when only 6 deeper FFNs are pruned (SM + ScFuFFNi₆), and further validated against the unpruned configuration (SM + ScFuFFN) in (d). The last row (e, f, g) shows entropy heatmaps for each configuration.

Figure 18 illustrates the effect of $\gamma$ on the headwise entropy distribution. The hyperparameter $\gamma$ employed to adjust the threshold margin in entropy regularization, defined as $\text{Tol}_{\text{margin}} = \gamma E_{\text{max}}$ (Algorithm1, line #3), effectively preventing over-regularization by ensuring that a sufficient fraction of heads maintains entropy values in the upper range $\frac{3\text{Max}}{4}$ to Max. As $\gamma$ increases from 0 to 0.15, only a small proportion of attention heads (0.7%) are situated in the highest entropy range. However, as $\gamma$ is increased beyond 0.15, the fraction of heads in this upper range starts increasing, reaching 2.08%, 3.47%,

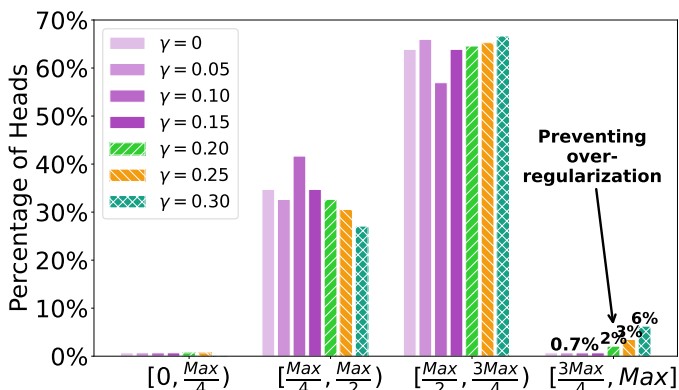

Figure 18: Headwise entropy distribution in the $\text{SM}(t) + \text{ScFuFFN}$ GPT-2 model ($L$=12, $H$=12, $d$=768) when entropy regularization is applied with varying threshold margin, controlled by hyperparameter $\gamma$.

and 6.25% at $\gamma$=0.20, 0.25, and 0.30, respectively. This fine-grained control on the population of attention heads in the higher entropy range highlights the ability of entropy regularization to prevent over-regularization and maintain the attention heads' diversity. We find that $\gamma$=0.2 yields slightly better performance in terms of lower perplexity compared to higher $\gamma$ values, and thus, we adopt this value in the final entropy regularization scheme.

To better understand the increase in the fraction of attention heads with higher $\gamma$, Figure 19 illustrates the layerwise entropy dynamics during training. Notably, at higher $\gamma$, the fraction of attention heads with higher entropy values increases, as indicated by the increases in the mean entropy of the early layers, which helps to prevent over-regularization and maintain heads' diversity.

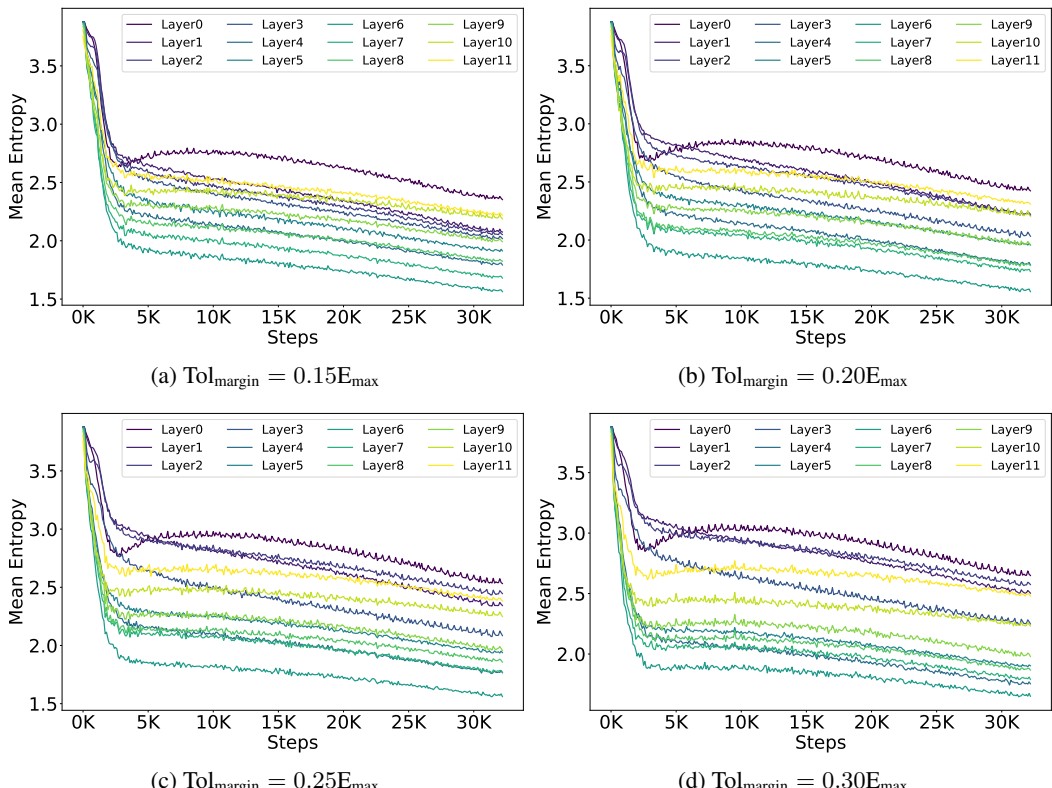

(a) $\text{Tol}_{\text{margin}} = 0.15 E_{\text{max}}$

(b) $\text{Tol}_{\text{margin}} = 0.20 E_{\text{max}}$

(c) $\text{Tol}_{\text{margin}} = 0.25 E_{\text{max}}$

(d) $\text{Tol}_{\text{margin}} = 0.30 E_{\text{max}}$

Figure 19: Layerwise entropy dynamics when entropy regularization is employed with increasing threshold margin, defined as $\text{Tol}_{\text{margin}} = \gamma E_{\text{max}}$ (see Algorithm1, line #3). At higher $\gamma$, the mean entropy of the early layers increases.

## E  FLOPs Computation for Inference

To generate one output token during inference, the model performs a forward pass over a sequence of length $L$ (the context size). Below, we detail the FLOPs required for both the feed-forward (FFN) and self-attention sub-blocks. We compute the *FLOPs per token per layer* as follows:

- **FFN FLOPs:** The FFN sub-block consists of two linear transformations, parameterized by $\mathbf{W}_{\text{in}}^{\text{ffn}} \in \mathbb{R}^{d \times 4d}$ and $\mathbf{W}_{\text{out}}^{\text{ffn}} \in \mathbb{R}^{4d \times d}$. Each layer contributes equally to the FLOPs count. The total FLOPs for the FFN can be expressed as:

$$\text{FFN FLOPs} = 2 \times 2 \times (d \times 4d) = 16d^2$$

  - The first factor of 2 accounts for the two linear layers, while the second factor of 2 arises because each dot product in a matrix-matrix multiplication involves two floating point operations—one multiplication and one addition (Performance, 2023).

- **Self-Attention FLOPs:** The breakdown for attention FLOPs is presented as follows:
  1. **Linear projections ($\mathbf{W}^Q$, $\mathbf{W}^K$, $\mathbf{W}^V$, and $\mathbf{W}^O$) FLOPs:** The input sequence of shape $\mathbb{R}^{T \times d}$ is linearly transformed using weights of shape $\mathbb{R}^{d \times d}$ across four linear layers (for queries, keys, values, and output projection). Thus, the total FLOPs for these operations are:

$$4 \times 2 \times T \times (d \times d) = 8Td^2$$

     Since we are interested in FLOPs per token, this simplifies to $8d^2$.
  2. **Attention Matrix ($\mathbf{QK}^T$) Computation:** The attention mechanism involves computing the dot product between the query matrix $\mathbf{Q} \in \mathbb{R}^{T \times d_k}$ and the transposed key matrix $\mathbf{K}^T \in \mathbb{R}^{d_k \times T}$. For each attention head, this operation results in:

$$2 \times T \times d_k \times T$$

     With $H$ heads, the total FLOPs for this step are:

$$2 \times H \times (T \times d_k \times T) = 2dT^2$$

     Hence, FLOPs per token simplifies to $2dT$.
  3. **Dot Product with V:** After calculating the attention weights, the values matrix $\mathbf{V} \in \mathbb{R}^{T \times d_k}$ is multiplied by the attention scores. Due to the masking in the upper triangular attention matrix (to enforce causality), only the lower triangular part of the matrix is involved in the computation. The number of FLOPs per head is:

$$2 \times d_k \times \frac{T(T+1)}{2}$$

     For $H$ heads, this totals to:

$$2 \times d \times \frac{T(T+1)}{2}$$

     Thus, the FLOPs per token for this step are:

$$d \times (T+1)$$

  Combining all components, the total FLOPs for self-attention per token is:

$$\text{Self-attention FLOPs} = 8d^2 + 2Td + d(T+1)$$

In summary, the FLOPs computation for one layer includes both the FFN and self-attention sub-blocks, yielding the following total per token:

$$\text{Total FLOPs per token per layer} = \underbrace{16d^2}_{\text{FFN}} + \underbrace{(8d^2 + 3Td + d)}_{\text{Self-attention}}$$

Total FLOPs with $L$ layers and $T$ tokens (context length) $= L \times T \times (24d^2 + 3Td + d)$

Note that the FFN FLOPs depend on the hidden dimension, a design hyperparameter typically set as a multiple of the model dimension $d$, which varies across LLM architectures, for example, $4d$ in GPT-2

(Radford et al., 2019; Brown et al., 2020) and Pythia(Biderman et al., 2023), $\frac{8d}{3}$ in LLaMA(Touvron et al., 2023), $3.5d$ in Mistral Jiang et al. (2023), and $8d$ in Gemma Team et al. (2024).

Now, we want to analyze which component, FFN or self-attention, dominates the total FLOPs count of a given architecture. For this, we solve the following inequality:

$$16d^2 > 8d^2 + 3dT + d \implies 8d^2 > d(3T + 1) \implies 8d - 1 > 3T \implies T < \frac{8d - 1}{3}$$

$$\implies T < \frac{8d}{3} \qquad \text{(To simplify, we approximate } 8d - 1 \approx 8d) \tag{13}$$

It is evident that in the shorter context length regime, where $T < \frac{8d}{3}$, the FFN FLOPs dominate the total FLOPs. Thus, for tasks with smaller context lengths, optimizing the FFN sub-block can lead to efficiency gains, while for larger contexts, self-attention FLOPs become more significant. Therefore, understanding this FLOPs distribution can guide efficient architectural design for private inference, depending on the expected context size.

### E.1 DISTRIBUTION OF FLOPS IN GPT-2 MODELS

Figure 20 illustrates the distribution of FLOPs between the attention and FFN sub-blocks across different context sizes for GPT-2 models, ranging from 128 to 8K tokens.

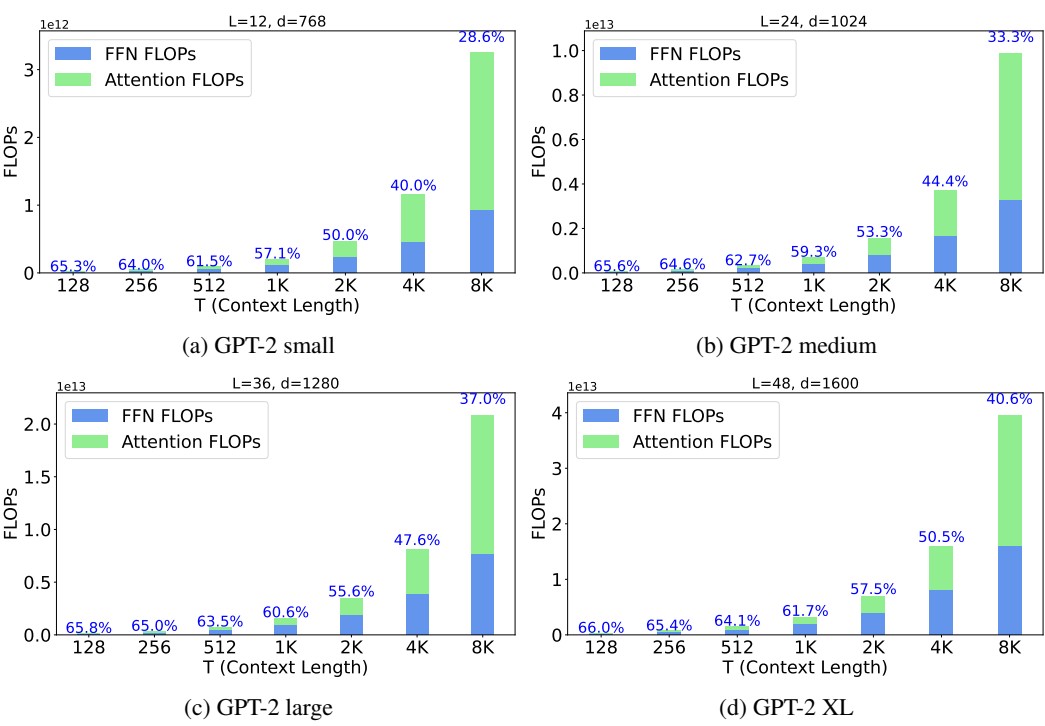

(a) GPT-2 small

(b) GPT-2 medium

(c) GPT-2 large

(d) GPT-2 XL

Figure 20: FLOPs breakdown in GPT-2 models for a single forward pass: Up to a context length of 2K, FFN operations are the primary contributors to FLOPs. Beyond 8K, attention operations start to dominate (Percentage on top of each bar represents the proportion of FFN FLOPs)

### E.2 DISTRIBUTION OF FLOPS IN PYTHIA MODELS

Figure 21 illustrates the distribution of FLOPs between the attention and FFN sub-blocks across different context sizes for Pythia models, ranging from 128 to 8K tokens.

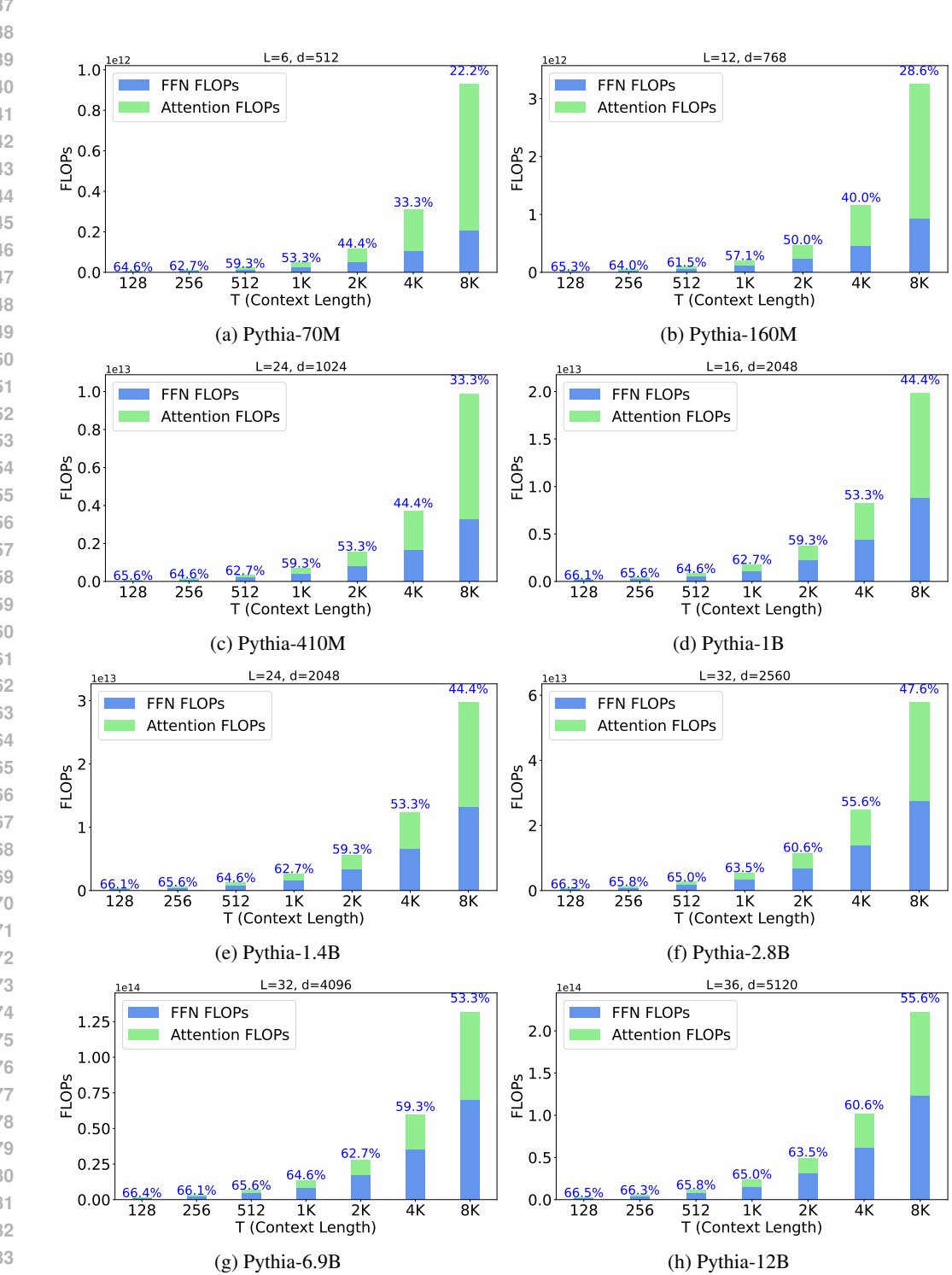

Figure 21: FLOPs breakdown in Pythia models for a single forward pass: Similar to GPT-2 models (see Figure 20), FLOPs are dominated by FFN operations up to a context length of 4K, except for smaller models where FFN operations dominate up to 2K (Percentage on top of each bar represents the proportion of FFN FLOPs).

# F    ADDITIONAL RELATED WORK

**The Pitfalls of LayerNorm in LLM** While LayerNorm is a critical source of non-linearity in both CNNs (Ni et al., 2024) and transformer-based models (Wu et al., 2024; Zhao et al., 2024; Joudaki et al., 2023), it causes several challenges in the transformer-based LLMs that extend well-beyond PI.

First, in PI with hybrid protocols, the inverse-square root computation poses significant challenges for its precise computation. Also, in HE-only PI, the polynomial approximation of LayerNorm is quite challenging because of the unusually wide range of its variance values Zimerman et al. (2024).

Furthermore, the trainable parameters in LayerNorm are shown to be associated with the outlier features in LLMs He et al. (2024); Bondarenko et al. (2023); Wei et al. (2022); Kovaleva et al. (2021), posing significant issues in LLM quantization. Specifically, the scaling parameters in LayerNorm amplify outlier features, which in turn makes low-precision training challenging Wei et al. (2022).

Moreover, LayerNorm introduces difficulties in mechanistic interpretability, as it tends to make the residual stream more complex and harder to analyze Nanda (2023).

Also, from the perspective of signal propagation theories, LayerNorm negatively impacts the trainability of LLMs He & Hofmann (2024); He et al. (2023).

**Entropy Regularization** Entropy regularization has been widely applied in various areas of machine learning. It has been used to penalize low entropy predictions (Pereyra et al., 2017) and to maximize predictions' entropy (Setlur et al., 2022). It has also been used to improve adversarial robustness (Jagatap et al., 2022), to avoid bad initializations and local minima (Miller et al., 1996), to optimize the layer-wise flow of information in deeper networks (Peer et al., 2022), to balance exploration-exploitation and promote action diversity in reinforcement learning (Wang et al., 2024; Ahmed et al., 2019; Lu & Van Roy, 2019; Neu et al., 2017; Mnih, 2016), and for domain generalization (Zhao et al., 2020).

**The Role of Nonlinearity in LLM** Understanding the role of nonlinearity in the transformer-based models is an emerging research area. Li et al. (2024) offer a theoretical analysis on the role of attention and FFN nonlinearity for in-context learning tasks. However, their work is limited to a simplified, one-layer model consisting of a single softmax-based self-attention head and a ReLU-based FFN. Nonetheless, Cheng et al. (2024) explore a broader range of nonlinear architectures and in-context learning tasks, demonstrating that the optimal activation function can vary depending on the specific function class the model is attempting to learn.

# G    DISCUSSION

## G.1    LINEAR VS. NON-LINEAR FFNS: PRIVILEGED BASIS AND ROTATIONAL INVARIANCE

In transformer-based architectures, there is a crucial distinction between linear and non-linear transformations, particularly when analyzing the *privileged basis* phenomenon (Elhage et al., 2023).

A *privileged basis* occurs when certain directions in the network's activation space are favored due to the network's architecture. Specifically, non-linear activations (e.g., ReLU, GELU) encourage features to align with specific directions in the hidden space, breaking the rotational invariance of the network. This alignment forces neurons to prioritize specific directions or features, leading to a basis that is more interpretable but inherently dependent on the chosen non-linear function.

**Non-linear FFN** In a conventional FFN, the architecture includes two linear transformations with a non-linear activation between them. This non-linearity breaks the rotational invariance of the hidden space, as the activation function behaves differently depending on the input direction. Mathematically, this can be described as:

$$\text{FFN}_{\text{non-linear}}(x) = \sigma(x\mathbf{W}_{\text{in}}^{\text{ffn}} + \mathbf{b}_1)\mathbf{W}_{\text{out}}^{\text{ffn}} + \mathbf{b}_2$$

Where $\sigma(\cdot)$ represents the elementwise non-linearity, which restricts the model from treating all directions in the hidden space equally, thereby establishing a *privileged basis*.

**Linear FFN** Removing the non-linearity from the FFN results in a purely *linear transformation*, which preserves rotational invariance in the hidden space. The linear FFN operates as:

$$\text{FFN}_{\text{linear}}(x) = (x\mathbf{W}_{\text{in}}^{\text{ffn}})\mathbf{W}_{\text{out}}^{\text{ffn}} + \mathbf{b}$$

Without the activation function, the representations remain invariant under rotations, meaning that no specific direction is favored over others. In a *linear FFN*, the model retains its flexibility in how it represents features, without forcing any particular alignment of neurons. In effect, the model operates in a *non-privileged basis*, allowing the representation to rotate freely in the vector space.

### G.2 Early vs. Deeper FFNs

Early FFNs in transformer models are particularly critical due to the presence of polysemantic neurons(Ferrando et al., 2024), which can respond to multiple, unrelated features simultaneously. These neurons enable early layers to detect broad and diverse patterns, such as linguistic structures, n-grams, and other foundational features of the input. This general-purpose functionality allows the early FFNs to play a vital role in the initial stages of context formation, capturing a wide range of information necessary for effective processing in later stages.

Additionally, research highlights that early layers play a pivotal role in the memorization and recall of factual information (Haviv et al., 2023; Meng et al., 2022). These layers not only capture general patterns but also store and retrieve memorized knowledge, promoting the predicted token to the top of the output distribution. This suggests that early FFNs are integral for accessing learned information, making them crucial for both context formation and recalling facts essential to the model's predictions.

In contrast, deeper FFNs become more specialized, focusing on refining the information extracted by the earlier layers. They process the data with a narrower scope, tailoring it for task-specific outputs. While these deeper layers are essential for fine-tuning the model's predictions, the early FFNs are key to generalizing over complex and varied input patterns, establishing the groundwork for the model's overall performance.

## H  Future Work

To further reduce non-linear operations in our Softmax-only architecture, off-the-shelf head pruning techniques Voita et al. (2019); Michel et al. (2019); Jo & Myaeng (2020); Ma et al. (2021); Li et al. (2022) can be applied on top of AERO. Another approach is to explore linear softmax operations. However, these linear softmax operations sometimes introduce additional normalization layers or complex activation functions in the FFN Zhang et al. (2024), which could increase the PI overheads, counteracting the intended efficiency improvements.

Additionally, incorporating weight and activation quantization Wu et al. (2023); Xiao et al. (2023); Dettmers & Zettlemoyer (2023) could further enhance the efficiency of private inference in our architecture.

Orthogonally, performance improvement techniques such as knowledge distillation (KD) can be employed to complement these optimizations Ko et al. (2024); Liang et al. (2023); Gu et al. (2023); Hsieh et al. (2023); Li et al. (2023b).

Looking ahead, scaling AERO to more complex and deeper LLMs can be achieved by strategically combining techniques such as weight normalization, spectral normalization, and FFN output scaling. These methods can be applied selectively, with different layers using different techniques—for instance, employing spectral normalization in early layers and FFN output scaling in deeper layers. This tailored approach could lead to better stability and efficiency in larger models.

## I  AERO Beyond Private Inference: Broader Impact and Promises

While AERO originally developed to enable efficient private inference in transformer-based language models by reducing cryptographic overheads, its architectural innovations and insights have

far-reaching implications beyond privacy-preserving computations. Its principled approach for architectural simplifications and techniques for maintaining model stability offer valuable insights into the broader field of language model design and optimization.

Here, we explore how AERO's principles can influence various facets of standard transformer architectures and their applications. These broader implications span from fundamental architectural design choices to practical deployment considerations, demonstrating how techniques developed for private inference can enhance the understanding and implementation of LLMs.

**Plaintext efficiency** Our finding that ReLU naturally emerges as the preferred activation function in LayerNorm-free architectures aligns well with plaintext efficiency goals. ReLU's ability to induce sparsity in activations accelerates the plaintext inference by reducing the data traffic between CPU and GPU Mirzadeh et al. (2024).

Furthermore, AERO introduces a systematic approach for reducing FLOPs by designing *linear FFNs*, which enable the fusion of linear layers into a single transformation and reduces FFN FLOPs by $8\times$. Moreover, AERO replaces deeper FFNs with identity connections without compromising training stability, resulting in *additional* FLOPs reductions. These optimizations are particularly effective for inference with smaller context lengths ($T < \frac{8d}{3}$), where FFN FLOPs dominates.

**Low-precision training and quantization for resource-constrained applications** The deployment of transformer models in resource-constrained environments is significantly hindered by outliers that complicate low-bitwidth quantization. These outliers emerge from two primary sources: softmax-based attention mechanism Hu et al. (2024b) and LayerNorm layers He et al. (2024); Wei et al. (2022); Kovaleva et al. (2021). LayerNorm-induced outliers are particularly *severe* as their scaling parameters amplify activation extremes, demanding higher bit-widths for accurate representation.

Our LayerNorm-free design inherently addresses the major source of outliers by completely removing the problematic scaling parameters. While empirical validation remains necessary, the LayerNorm-free architectural choices in the AERO framework point to promising directions for enabling low-precision training and quantization, where effective outlier management is critical.

**An information-theoretic approach for understanding the role of nonlinearity in LLMs** While prior work has examined the role of nonlinearities in in-context learning (Cheng et al., 2024; Li et al., 2024), AERO introduces an information-theoretic perspective to analyze their impact in LLMs. By employing Shannon's entropy as a quantitative measure, we characterize how architectural choices influence attention score distributions in MHA sub-block.

Our analysis reveals that nonlinearities in LLMs serve a *dual* purpose: they are essential not only for training stability but also for preserving attention head diversity, enabling the optimal utilization of MHA's representational capacity. Consequently, their removal can destabilize training, leading to entropy collapse in deeper layers, or result in the underutilization of MHA's representational capacity, as evidenced by entropic overload in early layers—or both.

**Insights into mechanistic interpretability and disentangling polysemantic neurons** By stripping away complex nonlinearities, such as GELU and LayerNorm, AERO makes it easier to dissect how individual neurons, attention mechanisms, and linear FFNs contribute to model behavior, thereby facilitating a more *granular* understanding of the internal dynamics of LLMs.

In standard transformer architectures, nonlinearities like GELU induce complex feature interactions, often resulting in polysemantic neurons—neurons that encode overlapping or unrelated information Gurnee et al. (2023); Mu & Andreas (2020). By simplifying the FFN to a purely linear structure, AERO could minimizes feature entanglement, fosters monosemantic neuron behavior, and produces a more disentangled and interpretable neuron space Pearce et al. (2024)

Moreover, while not explicitly explored in Gould et al. (2024); Kissane et al. (2024), there may be intriguing connections between the entropy regularization in AERO, which dynamically adjusts to head-specific behavior, and the insights offered by attention heads that inherently encode interpretable patterns. This suggests the possibility that entropy-guided, adaptive attention mechanisms could facilitate the emergence or identification of such interpretable structures.

**Parallels between entropy-guided attention and differential attention mechanism** In principle, our entropy regularization scheme shares fundamental similarities with the recently proposed Differential Transformer architecture Ye et al. (2024), despite their distinct implementations.

Just as the differential attention mechanism nullifies attention noise through contrastive learning, our entropy-guided attention mechanism can be tailored to achieves similar effects by penalizing attention dispersal across tokens. Both approaches effectively promote *sparse* attention patterns: Differential Transformer achieves this through computing differences between attention maps, while our entropy regularization specifically penalizes high-entropy attention distributions.

These connections demonstrate that promoting selective attention can be achieved either through architectural design or through careful entropy regularization, offering complementary approaches to the same underlying objective.

**Entropy-guided solution for attention sink and attention noise** The softmax operation in transformer-based attention mechanisms inherently assigns non-zero probabilities to all tokens due to its normalized exponential form. This leads to two key issues: disproportionate emphasis on specific tokens (attention sink) Xiao et al. (2024); Cancedda (2024); Gu et al. (2024); and non-zero scores for irrelevant tokens (attention noise), which may leads to hallucinations Ye et al. (2024), outlier activations Hu et al. (2024b), and inefficient use of model capacity.

While recent works Yin et al. (2024); Yu et al. (2024) have proposed various solutions, AERO introduces a principled approach with its adaptive entropy regularization to control attention distribution. By penalizing excessively high entropy values and incorporating learnable threshold parameters, AERO enables each attention head to adaptively determine its optimal degree of focus. This could prevent the over-diffusion of attention scores while *preserving* the mathematical properties of softmax.

**New scientific opportunities for understanding training instability** Recently, there has been significant interest in understanding training instability in transformer models Wortsman et al. (2024); Rybakov et al. (2024); Zhai et al. (2023). While these studies provide valuable insights, they predominantly focus on standard transformer architecture, leaving a critical question unaddressed: *How do architectural simplifications, such as the removal of non-linearities, impact training dynamics?*

Our work provides the first comprehensive investigation of how the systematic removal of nonlinearities impacts training stability, specifically through the lens of entropy dynamics, offering novel insights for understanding the interplay between model design and training dynamics.

Prior research has focused on instability factors such as the unbounded growth of attention logits Zhai et al. (2023); Dehghani et al. (2023). In contrast, our entropy-based analysis reveals a previously unexplored contributor to training stability: *linear FFNs*. Notably, we demonstrate that linear FFNs, especially in the early layers, are critical for preventing entropy collapse during training.

While our experiments use smaller models like GPT-2, the findings on training instability extend to larger models, as prior research Wortsman et al. (2024) shows similar patterns persist at scale.

**Static alternatives for training stability solutions** To address training instability, prior work has predominantly relied on LayerNorm applied to various parts of the network, such as QK-LayerNorm Dehghani et al. (2023); Wortsman et al. (2024); Rybakov et al. (2024). In contrast, our approach demonstrates the effectiveness of static alternatives like weight normalization and spectral normalization. These methods mitigate entropy collapse in deeper layers while avoiding the computational overhead and non-linear operations associated with LayerNorm during inference.

**Entropy-guided solutions for uncertainty estimation and mathematical reasoning** Recent advancements, such as the entropy-guided sampling method (e.g., the Entropix framework Team (2024)) and the discovery of entropy neurons which regulate uncertainty in next-token predictions Gurnee et al. (2024); Stolfo et al. (2024), highlight a significant shift toward entropy-guided LLMs solutions.

These approaches have the potential to address token-level challenges in mathematical reasoning tasks Langlais (2024). The recently introduced FrontierMath benchmarks Glazer et al. (2024), which have garnered significant attention from the research community, further emphasize the importance of improving the mathematical reasoning capabilities in LLMs.

The key insight from this line of research is that different mathematical operations may require varying levels of token selection certainty. For instance, basic arithmetic operations might benefit from more deterministic (low-entropy) token selection, whereas complex problem-solving tasks could thrive on more exploratory (high-entropy) patterns.

While AERO's current entropy regularization focuses on attention patterns, it could be adapted beyond attention patterns to also guide token selection during inference, similar to the *adaptive temperature* strategies that adjust model creativity based on logit entropy Veličković et al. (2024). Another potential approach would be to introduce controlled entropy pathways specifically for numerical operations, and different entropy thresholds for different types of mathematical tasks.

Furthermore, AERO's architecture could be augmented with specialized *reasoning tokens*, inspired by pause tokens designed for reasoning steps Langlais (2024). By extending entropy regularization to guide the model's behavior around these reasoning tokens, we could enable more structured and interpretable mathematical reasoning paths.

