# OpenReview forum: "AERO: Softmax-Only LLMs for Efficient Private Inference"
_ICLR.cc/2025/Conference — Submitted to ICLR 2025_

### Official Review · Reviewer_rkoD · 2024-11-02

**Soundness:** 3
**Presentation:** 4
**Contribution:** 3
**Rating:** 6
**Confidence:** 3

**Summary:**

This paper proposes the AERO framework, a new method to optimize large language models (LLMs) in resource-constrained contexts, such as private inference (PI), by systematically reducing non-linear operations until Softmax becomes the model’s sole source of non-linearity, thereby enhancing computational efficiency. The main contributions of this work are as follows:

1. In the Softmax-only model, the authors apply weight normalization and scaling techniques to the linear layers within the feed-forward network (FFN), effectively preventing training collapse. This shift from activation normalization to weight normalization avoids non-linear calculations during the inference phase, thereby improving inference efficiency.

2. In the Softmax-only architecture, the authors merge two linear layers in the FFN into a single linear layer, reducing the FFN’s FLOPs by 8x without compromising model performance. Additionally, they prune the deeper FFN layers, further enhancing computational efficiency.

3. To address issues of entropy overload and entropy collapse that are prone to occur during training in the Softmax-only architecture, the authors propose an entropy regularization method. By penalizing extreme entropy values during training, this method ensures a balanced entropy distribution across attention heads, maintaining model stability and performance.

**Strengths:**

The paper demonstrates a range of strengths in terms of originality, quality, clarity, and significance, as detailed below:

The AERO framework presents a novel approach by systematically reducing non-linear operations until Softmax becomes the model’s sole non-linearity, thereby enhancing computational efficiency. Its originality lies in the combination of several innovative techniques, including the use of weight normalization and scaling instead of traditional activation normalization, merging FFN layers to reduce FLOPs by 8x, and applying entropy regularization to address issues of entropy overload and entropy collapse during training.

The paper employs a rigorous research methodology, conducting experiments across multiple models (such as GPT-2 and Pythia) and various context lengths. Detailed ablation studies demonstrate the effectiveness of each component within the AERO framework, such as weight normalization and entropy regularization. These experimental results enhance the technical reliability of the paper and support its main conclusions.

The paper is well-structured, with clear explanations of its motivations, methods, and experimental results. Complex concepts, such as entropy regularization and the transition from activation normalization to weight normalization, are thoroughly explained, aiding readers in understanding the rationale and benefits behind these technical choices.

By improving the efficiency of LLMs in PI scenarios, the AERO framework addresses a critical issue in deploying LLMs in resource-constrained environments. This approach holds significant value in the field of privacy-preserving machine learning, potentially facilitating broader applications of LLMs in privacy-sensitive or resource-limited settings. The demonstrated reductions in FLOPs, communication costs, and inference latency contribute meaningfully to the field.

**Weaknesses:**

1. Lack of Detail in Entropy Regularization Implementation:
   While entropy regularization is an important part of the AERO framework, the paper does not provide sufficient detail on how the regularization thresholds (for addressing entropy overload and entropy collapse) are selected. A clearer explanation or additional experiments testing different threshold values could strengthen the understanding of this technique and its practical application.

2. Limited Comparison with Existing Non-linear Reduction Techniques:
   The paper primarily focuses on its unique approach but provides limited comparison with other existing methods that aim to reduce non-linear operations in LLMs. Incorporating experiments or discussions comparing AERO with similar frameworks could help highlight its relative strengths and weaknesses, making the contributions clearer.

3. Absence of Optimization with Specific Privacy-Preserving Techniques in Real-world PI Applications:
   Although the paper demonstrates AERO’s efficiency improvements in terms of FLOPs, communication cost, and inference latency, it lacks evaluations in real-world PI application scenarios, especially in conjunction with specific privacy-preserving techniques, such as multi-party computation or homomorphic encryption. Including benchmarks that evaluate AERO’s performance when combined with these privacy techniques could further showcase the method’s practicality and effectiveness in actual applications.

**Questions:**

1.Could the authors provide more detail on how the entropy regularization thresholds were selected for addressing entropy overload and entropy collapse? Were these values determined empirically, or was there a specific criterion used? Further insights here would help clarify the robustness of the regularization approach.

2.The paper highlights the novelty of reducing non-linear operations to a Softmax-only model, but could the authors provide additional comparisons with other existing methods beyond LayerNorm-free design? (Are there any more SOTA that aim to achieve similar goals?)

3.While weight normalization was shown to prevent training collapse, how does it affect the model's generalization ability in different contexts?

4.Could the authors elaborate on how AERO performs in real-world private inference scenarios, especially when integrated with specific privacy-preserving techniques like homomorphic encryption or multi-party computation?

5. Realizing the limited computational resource and time, the reviewer is still curious about AERO's performance on large models other than GPT-2 and Pythia-70M.

---

> ### Author Response · Authors · 2024-11-20
> **Author's Response (1/2)**
>
> Thank you for your comprehensive and thoughtful feedback. We are pleased that you recognized the originality of the AERO framework, including weight normalization as a static alternative for stabilizing training, FLOPs reduction, and entropy regularization.
>
> We greatly appreciate your acknowledgment of our rigorous methodology, comprehensive experiments, and clear presentation, as well as the practical significance of our work in advancing efficient LLMs for privacy-preserving applications. Below we addressed the following questions:
>
> >Lack of Detail in Entropy Regularization Implementation: While entropy regularization is an important part of the AERO framework, the paper does not provide sufficient detail on how the regularization thresholds (for addressing entropy overload and entropy collapse) are selected.
>
> Thank you for the insightful and important question. We greatly appreciate the reviewer’s interest in this aspect of our work. A detailed experimental analysis is provided in **Appendix D.8**.
>
> Specifically, Figure 18 in our revised manuscript illustrates the impact of varying thresholds ($\gamma$) on the headwise entropy distribution. As $\gamma$ increases from 0 to 0.15, only 0.7% of attention heads fall into the highest entropy range. However, beyond $\gamma = 0.15$, this fraction increases to 2.08%, 3.47%, and 6.25%  for $\gamma = 0.20$, 0.25, and 0.30, respectively. These results highlight how entropy regularization avoids over-regularization and effectively maintains attention head diversity.
>
>
> Based on these experiments, we find that $\gamma = 0.2$ achieves the best perplexity performance, which we have adopted as the final setting for our entropy regularization scheme.
>
> >Limited Comparison with Existing Non-linear Reduction Techniques: The paper primarily focuses on its unique approach but provides a limited comparison with other existing methods that aim to reduce non-linear operations in LLMs.
>
> We appreciate the reviewer's suggestion regarding comparisons with existing nonlinearity reduction techniques, beyond the LayerNorm-free design.
>
> To the best of our knowledge, the current SOTA LayerNorm-free model [1] represents the most relevant comparison point for our work, as it successfully preserves model performance while removing LayerNorm through architectural optimization.  While there are other approaches like polynomial approximation methods [2], they target different design goals -- approximating nonlinearities rather than eliminating them -- and often face fundamental limitations including *data-specific accuracy dependencies and narrow input ranges.*
>
> Our work takes a fundamentally different direction by systematically eliminating nonlinearities and introducing entropy regularization, achieving significant efficiency gains (4.23$\times$ communication and 1.94$\times$ latency reduction) while maintaining model stability. We believe these comprehensive comparisons against the current SOTA effectively demonstrate AERO's advantages in the specific context of efficient private inference.
>
> >Absence of Optimization with Specific Privacy-Preserving Techniques in Real-world PI Applications
>
> We appreciate the reviewer's interest in AERO's real-world privacy applications. We have indeed conducted comprehensive evaluations of AERO in practical private inference scenarios using both homomorphic encryption and secure multi-party computation protocols.  The client and server are simulated on two physically separate machines, each equipped with an AMD EPYC 7502 server (2.5 GHz, 32 cores, 256 GB RAM), operating in a WAN setting.
>
> As detailed in Section C.3 of our revised manuscript, we implement AERO using the BumbleBee's SecretFlow framework (Lu et al., NDSS 2025), which provides SOTA cryptographic protocol optimization for both linear operations (through efficient packing techniques) and the nonlinear (GELU, ReLU, LayerNorms, and Softmax) operations.
>
> Our experimental results in Tables 4-7 report actual end-to-end (including input embeddings and final output layers) private inference latency and communication costs measured in a realistic WAN setting (100Mbps bandwidth, 80ms latency). These results demonstrate that AERO achieves up to 4.23$\times$ communication and 1.94$\times$ latency reduction compared to the baseline.
>
>
> [1] Bobby He and Thomas Hofmann. Simplifying transformer blocks. ICLR 2024.
>
> [2] Itamar Zimerman, Moran Baruch, Nir Drucker, Gilad Ezov, Omri Soceanu, and Lior Wolf. Converting transformers to polynomial form for secure inference over homomorphic encryption. ICML 2024.

---

> ### Author Response · Authors · 2024-11-20
> **Author's Response (2/2)**
>
> >While weight normalization was shown to prevent training collapse, how does it affect the model's generalization ability in different contexts?
>
> We employ weight normalization [3] as a static alternative to LayerNorm, avoiding the computational cost of nonlinear operations during private inference. However, we have not fully explored its generalization capabilities across diverse contexts and tasks.
>
> As shown in Table 3, weight normalization, when employed in linear layers of FFN, underperforms compared to the other FFN-scaling method, and its effectiveness varies depending on where it is applied. Specifically, applying weight normalization to linear layers within the FFN achieves better results than applying it to layers in the attention module, when tested on the GPT-2 small model.
>
> Weight normalization is known to have nuanced effects on generalization, and their generalization benefits are not universal, depending heavily on the architecture and task [3]. Prior research [4] has reported that weight normalization can sometimes lead to underfitting if not combined appropriately with weight decay.
>
> >Realizing the limited computational resources and time, the reviewer is still curious about AERO's performance on large models other than GPT-2 and Pythia-70M.
>
> Thanks for raising an insightful question about AERO's potential scalability to larger models. We have addressed this thoroughly in our General Response (2/2).
>
> [3] Tim Salimans and Durk P Kingma. Weight normalization: A simple reparameterization to accelerate training of deep neural networks. NeurIPS 2016.
>
> [4] Xiang et al., Understanding the disharmony between weight normalization family and weight decay, AAAI 2020.

---

> > ### Author Response · Authors · 2024-11-25
> > **Discussion Period Nears End**
> >
> > Dear Reviewer rkoD,
> >
> > Thank you once again for your comprehensive and thoughtful feedback on our submission. As the discussion period nears its end, we are eager to know if our additional results and clarifications have adequately addressed your questions.
> >
> > We would sincerely appreciate any further perspectives or discussions you might have at this stage.
> >
> > Thank you for your time and engagement!
> >
> > Best regards,
> >
> > Authors

---

> > ### Comment · Reviewer_rkoD · 2024-12-03
> >
> > The reviewer thanks the authors for their response and keeps the score unchanged.

---

### Official Review · Reviewer_aqto · 2024-11-02

**Soundness:** 2
**Presentation:** 2
**Contribution:** 2
**Rating:** 6
**Confidence:** 4

**Summary:**

This paper studies the efficient private inference of LLM and proposes an algorithm using only Softmax nonlinear functions. The problem of privacy inference studied is of practical value. The proposed scheme can greatly reduce the nonlinear operation in LLM and has practical application value.

**Strengths:**

The proposed scheme can greatly reduce the nonlinear operation in LLM and has practical application value.

**Weaknesses:**

The introduction does not fully introduce the motivation of the research, and does not elaborate on whether the proposed algorithm can solve the challenges mentioned in the introduction. The experimental results lack more detailed evaluation of test accuracy and other indicators.

**Questions:**

1. The introduction of the paper emphasizes the importance and challenge of private inference. However, how the proposed AERO addresses these challenges is not fully described. Why reducing nonlinear operations can help LLMS perform private inferences needs to be explained in more detail.

2. How much gain and benefit will the proposed scheme bring to private inference? This requires further objective numerical evaluation.

3. The author describes in the limitation that this work mainly discusses the PPL performance of the model. What is the test performance on the actual NLP task? This needs to be explained in the experimental results section.

4. The content of the appendix is not in good shape, and there is no text introduction under many sub-headings, which makes it difficult for readers to obtain information from them.

---

> ### Author Response · Authors · 2024-11-20
> **Author's Response (1/2)**
>
> Thank you for acknowledging the practical significance of our proposed scheme in effectively reducing nonlinear operations in LLMs. We have addressed the questions raised by the reviewer, as follows:
>
> >How much gain and benefit will the proposed scheme bring to private inference? This requires further objective numerical evaluation.
>
> We would like to clarify that the paper already provides detailed numerical evaluations of the proposed scheme's benefits to private inference. Specifically:
>
> 1. *Communication and Latency Gains:* As reported in the Results section (Table 4 and Figure 1), the proposed AERO framework achieves up to **4.23$\times$** reduction in communication overhead and **1.94$\times$** reduction in latency compared to the baseline models during private inference. These improvements are evaluated in realistic private inference scenarios using models like GPT-2. The client and server are simulated on two physically separate machines, each equipped with an AMD EPYC 7502 server (2.5 GHz, 32 cores, 256 GB RAM), operating in a WAN setting.
>
> 2. *Comprehensive Benchmarks:* The results have been validated across various context lengths (128, 256, and 512 tokens) and model depths (12 and 18 layers), ensuring the robustness and applicability of the framework (Section 5, Table 7).
>
>
> We believe these evaluations provide a comprehensive and objective analysis of the gains achieved by AERO for private inference.
>
>
> >The author describes in the limitation that this work mainly discusses the PPL performance of the model. What is the test performance on the actual NLP task? This needs to be explained in the experimental results section.
>
> Thank you for highlighting this important point. Our primary focus in this work was to analyze the architectural implications of removing LayerNorm and FFN activation functions, particularly within the context of pre-training. In particular, to examine the influence of nonlinearities on LLM internal dynamics and model behavior to design nonlinearity-efficient LLMs for private inference.
>
> Perplexity (PPL) was chosen as the primary evaluation metric because it provides a direct measure of the model's ability in sequence modeling tasks and reflects the quality of the learned representations at this stage. Nonetheless, we acknowledge the importance of evaluating test performance on downstream NLP tasks to fully assess the utility of the proposed architecture.
>
> However, including such evaluations was beyond the scope of this paper due to resource and time constraints. Instead, we focused on presenting a rigorous analysis of the architectural modifications and their impact on pre-training dynamics, including entropy-based insights that are novel to this work.
>
> That said, we are confident that the improvements in PPL observed in our experiments are indicative of better pre-trained representations, which often translate into stronger performance on downstream tasks. As future work, we plan to conduct a comprehensive evaluation of the proposed architecture on popular NLP benchmarks to quantify its downstream effectiveness.
>
> >The content of the appendix is not in good shape, and there is no text introduction under many sub-headings, which makes it difficult for readers to obtain information from them.
>
> Thank you for the careful review of the Appendix. We sincerely appreciate the reviewer's time and effort in examining it thoroughly. The suggested changes have been incorporated into our revised manuscript.

---

> ### Author Response · Authors · 2024-11-20
> **Author's Response (2/2)**
>
> >The introduction of the paper emphasizes the importance and challenge of private inference. However, how the proposed AERO addresses these challenges is not fully described. Why reducing nonlinear operations can help LLMS perform private inferences needs to be explained in more detail.
>
> Thank you for this feedback. We agree this connection could be better emphasized. The cost disparity between linear and nonlinear operations in privacy-preserving LLM inference arises from the efficiency of their respective cryptographic implementations.
>
> Linear operations, such as matrix multiplications and additions, are highly optimized for privacy-preserving computation. Using homomorphic encryption (HE), these operations: (1) *leverage SIMD* (Single Instruction Multiple Data) through efficient packing techniques, (2) *allow batch processing* to significantly increase the throughput, (3) *incur minimal communication overhead* (often require a single-round), (4) and *achieve substantial communication reduction* through optimized packing techniques which yield up to **80 to 90% reduction** in communication costs [1].
>
> In contrast, nonlinear operations such as GELU and LayerNorm require secure multi-party computation (MPC), which involves **interactive protocols with substantial communication and computational costs**. Nonlinearities rely on cryptographic mechanisms like secure comparisons, oblivious transfer (OT), and polynomial evaluations (e.g., for GELU), which are inherently more expensive. For instance, a single GELU activation in a BERT-base model requires approximately $3.9 \times 10^6$ point-wise operations [1], each involving multiple secure multiplications and communication rounds, typically adding **1 to 2 KB per operation**
>
>
> More concretely, recent work CipherGPT [2] has quantitatively shown that non-linear operations are the major bottleneck in private inference -- GELU and LayerNorm together account for **49%** of latency costs and **59%** of communication costs, while linear operations like MatMul only contribute 19% to latency.
>
> Thus, our work AERO provides a systematic architectural solution to address these bottlenecks rather than focusing on cryptographic protocol optimizations. AERO systematically reduces these bottlenecks by 1) Eliminating LayerNorm, removing its inherent complexity for nonlinear computation at inference; 2) Replacing GELU with ReLU, a simpler activation that is cryptographically cheaper; and 3) Further simplifying FFNs in softmax-only configuration.
>
> By minimizing these bottlenecks, AERO enables faster and more communication-efficient PI, directly addressing the challenges identified in the introduction.
>
> [1] Lu et al., "Bumblebee: Secure two-party inference framework for large transformers." NDSS 2025
>
> [2] Hou et al., "CipherGPT: Secure Two-Party GPT Inference,"  Cryptology ePrint Archive, 2023.

---

> > ### Author Response · Authors · 2024-11-25
> > **Discussion Period Nears End**
> >
> > Dear Reviewer aqto,
> >
> > Thank you once again for your comprehensive and thoughtful feedback on our submission. As the discussion period nears its end, we are eager to know if our additional results and clarifications have adequately addressed your questions.
> >
> > We would sincerely appreciate any further perspectives or discussions you might have at this stage.
> >
> > Thank you for your time and engagement!
> >
> > Best regards,
> >
> > Authors

---

> > > ### Comment · Reviewer_aqto · 2024-12-01
> > > **Thank you for the response**
> > >
> > > The authors addressed my concerns.  I will keep my scores unchanged.

---

### Official Review · Reviewer_KStG · 2024-11-04

**Soundness:** 2
**Presentation:** 3
**Contribution:** 3
**Rating:** 6
**Confidence:** 5

**Summary:**

This paper presents an optimization framework to make large language models more efficient for private inference by minimizing non-linear operations such as LayerNorm and GELU. The proposed architecture, AERO, includes a Softmax-only model that reduces both communication and latency overheads. A novel entropy regularization technique is introduced to prevent training instabilities and entropic overload in the model.

**Strengths:**

1. This paper highlights an important research area—private inference (PI)—and extends the study of PI for LLMs by providing valuable insights into the impact of LayerNorm, a topic that has not been extensively explored before.

2. The insights into entropic overload and the proposed solution, entropy regularization, are both novel contributions.

**Weaknesses:**

1. Replacing GeLU with ReLU has been proposed in the work [1]. Despite this paper proposes insightful analysis, this method is not totally novel.
[1] Dake Chen, Yuke Zhang, Souvik Kundu, Chenghao Li, and Peter A Beerel. Rna-vit: Reduceddimension approximate normalized attention vision transformers for latency efficient private inference. In IEEE/ACM International Conference on Computer Aided Design (ICCAD), 2023.

2. Although AERO achieves notable reductions in communication overhead and latency, there is a trade-off in terms of higher perplexity compared to baseline models. As Iron [2] provides efficient private inference protocols for layernorm, I would suggest the authors to compare the PPL and PI savings with Iron.

[2] Meng Hao, Hongwei Li, Hanxiao Chen, Pengzhi Xing, Guowen Xu, and Tianwei Zhang. "Iron: Private inference on transformers." Advances in neural information processing systems 35 (2022): 15718-15731.

3. The acronym FFN is first introduced in line 70, but its full name, Feed-Forward Network, does not appear until line 108. This slight inconsistency in placement could lead to some confusion for readers.

**Questions:**

The proposed AERO framework and experiments primarily focus on models with fewer than 1B parameters, limiting the insights into its performance and scalability on larger LLMs commonly used in industry. This might restrict the applicability of the findings to more demanding real-world scenarios. What are the authors' perspective on how AERO might perform with larger models and whether they anticipate similar benefits or new challenges at a larger scale?

---

> ### Author Response · Authors · 2024-11-20
> **Author's  Response**
>
> Thank you for appreciating our focus on LLM private inference and our insights into the impact of LayerNorm, a relatively unexplored topic. We are pleased that you found our contributions on entropic overload and entropy regularization both novel and valuable.  We hope that our comments below address the reviewer's questions:
>
> >Replacing GeLU with ReLU has been proposed in the work [1]. Despite this paper proposes insightful analysis, this method is not totally novel.
>
> Thank you for raising this point and giving us the opportunity to clarify the novelty of our ReLU-related findings. While replacing GeLU with ReLU for plaintext and private inference efficiency is well-known, our key contribution lies in analyzing activation functions in **LayerNorm-free architectures**, where we find that the *geometrical* properties of ReLU play a crucial role in the absence of LayerNorm.
>
> Previous works [1] focus on ViT architectures that retain normalization layers and primarily evaluate on image classification tasks (e.g., CIFAR-10, CIFAR-100, Tiny-ImageNet), and conclude that LeakyReLU provides the best accuracy-to-latency (A2L) trade-off (Table 1 in [1]).
>
> However, in the context of LayerNorm-free LLMs, our findings diverge significantly: ReLU consistently outperforms other activations, including LeakyReLU. We demonstrate that ReLU's zero negative slope emerges as a natural preference in LN-free architectures, **contrary** to the preference for LeakyReLU in ViTs with normalization layers. This conclusion is supported by our learnable slope experiments (Figure 4), where slope values naturally converge to zero during training in LN-free setups.
>
> Moreover, our entropy-based analysis offers new insights into why ReLU performs better in LN-free settings -- it helps prevent entropic overload in early layers (Figure 5), a phenomenon not explored in [1] or other prior works.
>
> >Although AERO achieves notable reductions in communication overhead and latency, there is a trade-off in terms of higher perplexity compared to baseline models. As Iron [2] provides efficient private inference protocols for layernorm, I would suggest the authors to compare the PPL and PI savings with Iron.
>
> Thank you for this insightful comment. We would like to address the comparison with IRON [2] and clarify the following key points:
>
> 1. *Cryptographic Protocol Efficiency:* We use BumbleBee [3] as the underlying cryptographic protocol, which significantly outperforms IRON -- **92% less communication and is 13$\times$ faster** in end-to-end inference time for BERT models -- representing the current state-of-the-art.
>
> 2. *Pre-trained vs. Training from Scratch:* IRON uses pre-trained models, which naturally preserves accuracy since the architecture remains unchanged. In contrast, we train from scratch with architectural modifications specifically designed for PI efficiency.
>
> 3. *Fundamentally Different Contributions:* IRON focuses on cryptographic protocol optimization while preserving the original architecture. In contrast, AERO introduces architectural innovations specifically designed for PI efficiency by (a) Systematically analyzing and removing non-linearities, (b) Introducing entropy regularization to maintain model stability, and c) Reducing FLOPs through targeted architectural refinements.
>
>
> Therefore, a direct comparison may not be appropriate as the approaches are complementary rather than competing.
>
> >The acronym FFN is first introduced in line 70, but its full name, Feed-Forward Network, does not appear until line 108.
>
> We appreciate the reviewer's attentive reading and for pointing out the discrepancy. We have rectified this in the revised version of our manuscript.
>
> >What are the authors' perspective on how AERO might perform with larger models and whether they anticipate similar benefits or new challenges at a larger scale?
>
> Thank you for this thoughtful question. The scalability of AERO to larger models is indeed an important and exciting direction for future research. We have thoroughly addressed this in our General Response (2/2)
>
>
> [1] Dake Chen, Yuke Zhang, Souvik Kundu, Chenghao Li, and Peter A Beerel. "Rna-vit: Reduced dimension approximate normalized attention vision transformers for latency efficient private inference." ICCAD 2023.
>
> [2] Meng Hao, Hongwei Li, Hanxiao Chen, Pengzhi Xing, Guowen Xu, and Tianwei Zhang. "Iron: Private inference on transformers." NeurIPS 2022.
>
> [3] Wen-jie Lu, Zhicong Huang, Zhen Gu, Jingyu Li, Jian Liu, Kui Ren, Cheng Hong, Tao Wei, and WenGuang Chen. "Bumblebee: Secure two-party inference framework for large transformers." NDSS 2025

---

> > ### Author Response · Authors · 2024-11-25
> > **Discussion Period Nears End**
> >
> > Dear Reviewer KStG,
> >
> > Thank you once again for your comprehensive and thoughtful feedback on our submission. As the discussion period nears its end, we are eager to know if our additional results and clarifications have adequately addressed your questions.
> >
> > We would sincerely appreciate any further perspectives or discussions you might have at this stage.
> >
> > Thank you for your time and engagement!
> >
> > Best regards,
> >
> > Authors

---

> > > ### Comment · Reviewer_KStG · 2024-11-29
> > > **Thank you for the response**
> > >
> > > I would like to thank the authors for their response, which has resolved my questions. I will keep my scores unchanged.

---

### Author Response · Authors · 2024-11-20
**General Response (1/2)**

We sincerely thank all reviewers for their comprehensive, thoughtful, and insightful feedback, as well as their encouraging evaluations. We deeply appreciate the time and effort invested in carefully reading and evaluating our work. We are particularly pleased that the reviewers recognized the novelty of our contributions and acknowledged the potential impact of our insights in advancing LLM private inference.

In response to the reviewers' thoughtful suggestions, we have revised our manuscript to include several new additions in the Appendix (**highlighted in blue** in the revised version):

**C.3: Cryptographic Protocols for Linear and Nonlinear Operations:** Detailed analysis of cryptographic protocols for linear (FLOPs) and nonlinear operations (GELU, ReLU, LayerNorm, Softmax) used for LLM private inference.

**D.7: Training Dynamics in Softmax-only Models with Fewer FFNs:** Analyzes how pruning deeper FFNs in Softmax-only architecture affects training stability, demonstrated as entropy collapses in deeper layers (Figure 16 and Figure 17).

**D.8: Mitigating Over-Regularization with an Appropriate Threshold Margin:** Describes the strategies for selecting threshold margins to avoid over-regularization in our entropy-based regularization implementations (Figure 18 and Figure 19).

**I: AERO Beyond Private Inference: Broader Impact and Promises**  Discusses the broader implications of the principled approach to architectural simplification in AERO, as well as the entropy regularization techniques.

Additionally, as suggested by **Reviewer aqto**, we have added descriptions to contextualize and understand the experimental results presented in the Appendix. We have also updated the Related Work section (Appendix Section F) to include the prior work on the  nonlinearities's role in LLM

We would be glad to receive further feedback on the new results and welcome any perspectives on the discussion presented in Appendix I.

We sincerely hope that our revisions address all major concerns and questions raised by the reviewers. We remain eager to discuss and address any outstanding issues and further improve the work based on the reviewers' guidance.

---

> ### Author Response · Authors · 2024-11-20
> **General Response (2/2)**
>
> Here, we would like to address a thoughtful inquiry raised by Reviewer **KStG** and Reviewer **rkoD** regarding scaling AERO’s principles to larger models (>1B parameters) and the benefits of our proposed entropy regularization technique.
>
> **Response:**
> Our focus on smaller models is motivated by their relevance in private inference research (e.g., Zhang et al., NDSS 2025; Lu et al., NDSS 2025; Zimerman et al., ICML 2024), where computational and communication constraints pose significant challenges for larger models. These constraints make it critical to first establish efficient processing strategies for smaller models before scaling.
>
> While our experiments focus on models with fewer than 1B parameters due to computational constraints, we have strong theoretical and empirical reasons to believe that AERO’s benefits will extend to larger-scale models.
>
> Recent research [1] indicates that training instability patterns in transformer architectures are **consistent across model scales**. Our entropy-based analysis highlights fundamental architectural insights into the role of non-linearities and the critical contribution of early linear FFNs in preventing entropy collapse--principles *inherent* to the transformer architecture rather than specific to model size.
>
> That said, we acknowledge that scaling up to larger models may introduce new challenges, such as increased sensitivity to architectural changes and the need to carefully tune the entropy regularization hyper-parameters to leverage their benefits while maintaining training stability. We plan to validate AERO's performance on larger models in future work to further confirm its scalability and robustness.
>
> In summary, while our current experiments are constrained to smaller models, we believe AERO provides foundational insights that will remain relevant and valuable as private inference applications scale to larger models. We look forward to exploring these directions in future studies.
>
> 1. Wortsman et al, Small-scale proxies for large-scale Transformer training instabilities, ICLR 2024

---

### Meta-Review · Area_Chair_pgqA · 2024-12-22

**Metareview:**

The reviewers had reasonable arguments in favour of the paper and against it: on one hand they appreciated the approach of reducing non-linear operations and the structure of the work, on the other they had concerns with (a) higher perplexity and (b) limited comparison with prior work. All reviewers responded to the author feedback with very short responses indicating that they would not change their scores. No reviewers engaged in further discussion of the paper. After going through the paper and the discussion I have decided to vote to reject based on the above issues. Specifically, for (a) a reviewer pointed out that AERO had higher perplexities than baselines, and proposed an additional baseline. The authors argued that the baseline was out of scope as it has been shown to be much slower than Bumblebee. I agree with the authors on this. However, this sidesteps the concern about increased perplexity. Other tasks in the appendix show increased perplexity as well, and while this is to be expected it highlights a key downside to the approach: there is no way to tune the perplexity/communication/latency trade-off. If, after Step 4 of Figure 8, one ends up with a model that has a perplexity that is too high, they have no alternative. Whereas, methods that are aimed at approximations of non-linearities often have ways to trade-off the accuracy of the approximation with the communication/latency of the resulting network. This poses a serious threat to the applicability of the method. For (b), reviewers wanted to see comparisons with other methods for PI. While the above baseline was convincingly argued to be out of scope by the authors, the authors also argued that polynomial approximation methods have different design goals and face limitations including data-specific accuracy dependencies and narrow input ranges and so they are also out of scope. I disagree on this point: this only means that these methods cannot be applied on certain tasks, but this can be indicated in a results section. There is nothing stopping the authors from comparing against these methods. Given all of the above, I believe this work should be rejected at this time. Once these things and other issues mentioned in the reviews are addressed in an updated version, the work will be much improved.

**Additional Comments On Reviewer Discussion:**

See the above meta review for details on this. Further, I disregarded the review of Reviewer aqto as it was extremely short and clear that they had not read much of the paper. I would not recommend them as a reviewer for future ICLR conferences.

---

### Decision · Program_Chairs · 2025-01-22

Reject